# TEST-TIME EXPLORATION IN UNKNOWN ENVIRONMENTS

## ABSTRACT

With the continuous advancement of Large Language Models (LLMs), intelligent agents are becoming increasingly vital. However, these agents often fail in environments governed by implicit rules—hidden constraints that cannot be observed directly and must be inferred through interaction. This causes agents to fall into repetitive trial-and-error loops, ultimately leading to task failure. To address this challenge, we propose **Test-Time Exploration (TTExplore)**, a framework where a thinker component analyzes interaction history to infer these implicit rules and guide an actor. As training a thinker is challenged due to sparse task rewards, we introduce a novel training pipeline for stable reinforcement learning by incorporating techniques such as task decomposition and difficulty filtering. Using this pipeline, we train a specialized 7B model, **Exp-Thinker**. Evaluated on five text-based embodied tasks, TTExplore with our trained Exp-Thinker significantly improves baseline agent scores by an average of 14-19 points, demonstrating the effectiveness of explicitly reasoning about implicit rules.

## 1 INTRODUCTION

Large language model (LLM) based agents have demonstrated remarkable capabilities in assisting humans across diverse domains, including deep research (Jin et al., 2025; Song et al., 2025a;b; Chen et al., 2025), GUI navigation (Qin et al., 2025; Wu et al., 2024; Yang et al., 2024b; Hong et al., 2024), embodied AI (Chang et al., 2024; Wang et al., 2025; Song et al., 2024b) and etc. These agents function through iterative interactions with the environment, involving state observation, action generation, and feedback understanding to accomplish tasks. As agent applications expand to more complex real-world scenarios, improving the abilities of agents has become a critical research challenge. Mainstream researches focus on enhancing agents' inherent general capabilities, such as planning and reasoning, through prompt engineering (Yao et al., 2023; Shinn et al., 2023) or fine-tuned methods (Yin et al., 2023; Chen et al., 2024; Xi et al., 2024; Hu et al., 2024).

While existing approaches have achieved notable progress, they share a fundamental limitation: **agents struggle when operating in environments governed by implicit rules**. We distinguish between two forms of environmental knowledge: (i) explicit information, which is directly observable, and (ii) implicit rules, which remain hidden and require reasoning to uncover. In unknown environments, agents can typically interpret explicit information and adapt their actions accordingly, but they often fail to identify and reason about implicit rules. Although prior methods, such as ReAct (Yao et al., 2023) and Reflection (Shinn et al., 2023), enable agents to plan or reflect on erroneous actions, they lack a dedicated mechanism for systematically exploring the implicit rules. This weakness is especially evident in tasks that depend on unobservable, environment-specific knowledge. Without such understanding, agents tend to fall into local exploration loops or repetitive trial-and-error behaviors, ultimately resulting in task failure.

To illustrate this point, consider the Alfworld task (Shridhar et al., 2020) shown in Figure 1, where agents must navigate household scenarios with implicit rules such as "must be in front of an object before interacting with it" and "cannot hold two objects at the same time". When an agent without knowledge of these rules attempts to interact with objects, it frequently receives uninformative feedback like "Nothing happened". That leads to local exploration traps where agents repeatedly attempt similar invalid actions or inefficient trial-and-error. For such tasks, the main bottleneck is not the general abilities of agents, but their ability to explore and understand usable environmental rules.

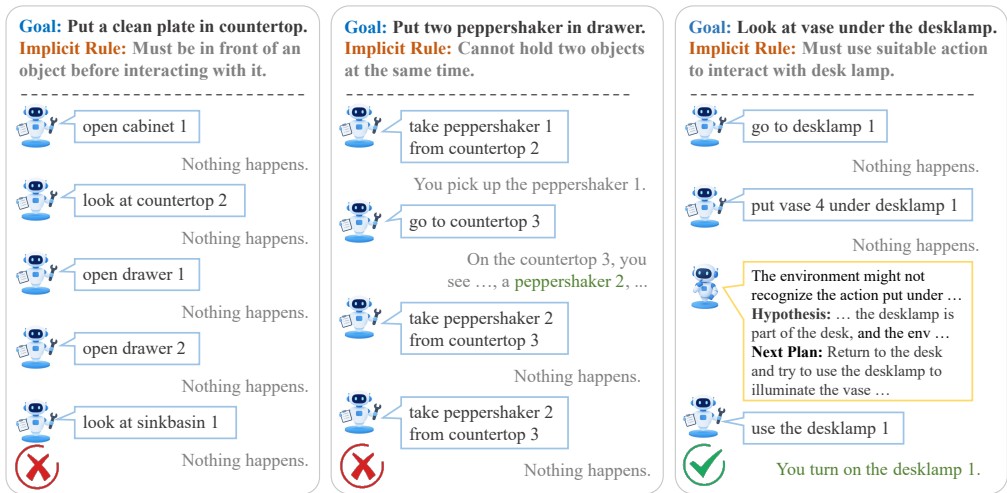

Figure 1: Three task examples of the Alfworld environment. We annotate the "implicit rule" of each task, which are deduced by humans but the agents can not see.

Without this ability, even the most powerful agent will perform tasks based on flawed assumptions, leading to inevitable failure.

To address this challenge, we propose **Test-Time Exploration (TTExplore)**, a framework designed to enable agents to explore and discover implicit environmental rules during test-time interaction. The framework comprises two roles: an actor that generates ReAct-style actions (Yao et al., 2023) and a thinker that performs deep thinking to reason the implicit rules and guide exploration. The thinker monitors task execution by analyzing the actor's actions and environmental feedback. When recent interactions include failed actions, it summarizes the task trajectory, infers implicit environmental rules, and proposes revised plans, as illustrated in the right panel of Figure 1. Throughout task execution, multiple deep thinking steps can be interleaved to regulate the actor's behavior and facilitate task completion.

The thinker role in our framework is central to exploring environmental knowledge, making its deep thinking ability particularly important. However, improving a model's thinking ability is highly challenging because we lack direct reward signals to evaluate the quality of its thoughts. A natural idea is to use final task performance (success or failure) as an indirect and sparse reward signal to optimize the thinker role via Reinforcement Learning (RL), such as the GRPO (Shao et al., 2024) algorithm. Yet, directly relying on sparse task rewards for RL training is unstable and inefficient. To overcome this issue, we construct a more stable training environment with a higher signal-to-noise ratio through methods such as task decomposition, difficulty filtering, and sampling trajectories with only a single deep thought. These designs allow us to more effectively attribute task rewards to key thoughts, enabling us to successfully train a 7B thinker model, **Exp-Thinker**.

We evaluate our TTExplore framework with the specially trained thinker model Exp-Thinker on five text-based embodied tasks from Agentboard (Chang et al., 2024). Experimental results demonstrate that our methods significantly improves agent performance on these environmental interaction tasks. Compared with the baseline models LLaMA3-8B and Qwen2.5-7B, our methods can improve their average performance by approximately 19 and 14 points, respectively. Furthermore, our method is also effective when combined with the agent training method. Even when the actor model in our framework is well-trained on in-domain tasks, the thinker model can still enhance performance on out-of-domain tasks.

## 2 TASK FORMULATION

Agent tasks that involve multiple rounds of interaction with the environment can be formalized as a Partially Observable Markov Decision Process (POMDP), defined by the tuple $(\mathcal{U}, \mathcal{S}, \mathcal{A}, \mathcal{O}, \mathcal{T})$. Here, $\mathcal{U}$ denotes the instruction space, $\mathcal{S}$ denotes the environment state space, $\mathcal{A}$ denotes the action

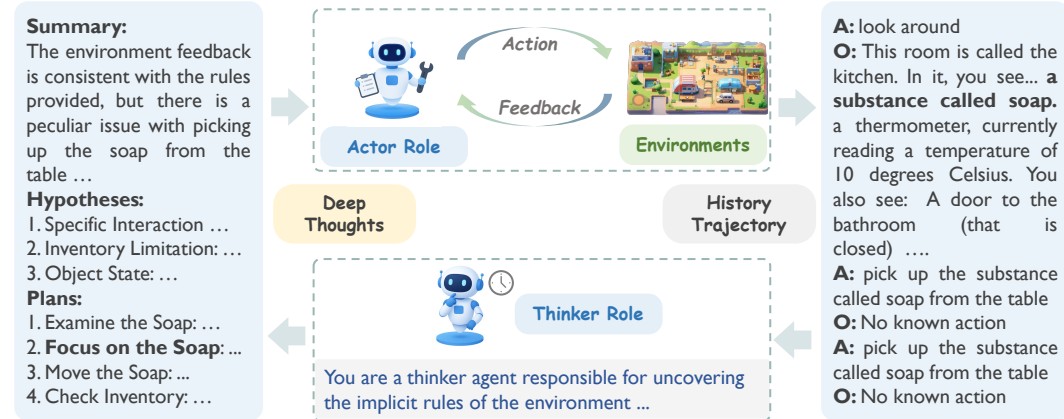

**Summary:**
The environment feedback is consistent with the rules provided, but there is a peculiar issue with picking up the soap from the table …

**Hypotheses:**
1. Specific Interaction …
2. Inventory Limitation: …
3. Object State: …

**Plans:**
1. Examine the Soap: …
2. **Focus on the Soap:** …
3. Move the Soap: …
4. Check Inventory: …

Deep Thoughts

History Trajectory

Thinker Role

You are a thinker agent responsible for uncovering the implicit rules of the environment …

Action

Feedback

Actor Role

Environments

**A:** look around
**O:** This room is called the kitchen. In it, you see... **a substance called soap.** a thermometer, currently reading a temperature of 10 degrees Celsius. You also see: A door to the bathroom (that is closed) ….
**A:** pick up the substance called soap from the table
**O:** No known action
**A:** pick up the substance called soap from the table
**O:** No known action

Figure 2: The actor role and the thinker role in our TTExplore framework.

space, and $\mathcal{O}$ denotes the observation space. The transition function $\mathcal{T} : \mathcal{S} \times \mathcal{A} \rightarrow \mathcal{S}$ specifies how the environment state evolves in response to an action.

For a given task, the agent receives an instruction $u \in \mathcal{U}$, starts from an initial environment state $s_0 \in \mathcal{S}$, and observes an initial partial observation $o_0 \in \mathcal{O}$. At each step $t$, given the previous state $s_{t-1}$ and an action $a_t$, the environment transitions to a new state $s_t$ and produces a partial observation $o_t$. Since the agent $\pi$ cannot access the full environment state $s_t$, it must instead rely on the sequence of historical observations and actions. Typically, an agent $\pi_\theta$ parameterized by $\theta$ predicts the next action $a_{t+1}$ conditioned on the trajectory observed so far, as shown in Eq. 1:

$$a_{t+1} \sim \pi_\theta(\cdot \mid u, o_0, a_1, o_1, a_2, o_2, \dots, a_t, o_t). \tag{1}$$

This interaction loop continues until the task is completed or a maximum step limit is reached. Finally, a reward $r$ is provided to evaluate task performance over the entire trajectory $traj$. Crucially, in many real-world complex tasks, the observations $\mathcal{O}$ do not explicitly reveal the underlying environment rules. The central challenge for agents is to infer these implicit rules through interaction with the environment, rather than relying solely on surface-level information.

## 3 METHOD

To enable agents to explore and understand implicit environmental rules during test time, we propose the **Test-Time Exploration (TTExplore)** framework. At its core is the *thinker* role, a key component that drives effective environment exploration. In this section, we first introduce the overall framework, and then describe our approach to training a specialized thinker model.

### 3.1 TEST-TIME EXPLORATION FRAMEWORK

The core principle of our TTExplore framework is to decouple low-level action execution from high-level strategic reasoning. As shown in Figure 2, We instantiate this through two distinct roles: an actor $\pi^{actor}$ and a thinker $\pi^{thinker}$. The actor serves as the primary interaction agent, generating ReAct-style outputs $a_t$ at each step $t$, which consist of a short thought followed by an executable action. In contrast, the thinker functions as a meta-level reasoning agent. It is not invoked at every step; instead, it is triggered when the agent risks getting stuck or requires a deeper understanding of the environment's rules. For simplicity, we activate the thinker at a fixed frequency of $n$ steps in our experiments. Once invoked, the thinker processes the entire trajectory history and produces a deep thought $d_i$, as shown in Eq. 2.

$$d_i \sim \pi_\theta^{thinker}(\cdot|u, o_o, a_1, o_1, a_2, o_2, \dots, a_i, o_i). \tag{2}$$

This deep thought typically contains three components: a summary of the current progress, a hypothesis about the latent environmental rules underlying recent failures, and a revised plan for future

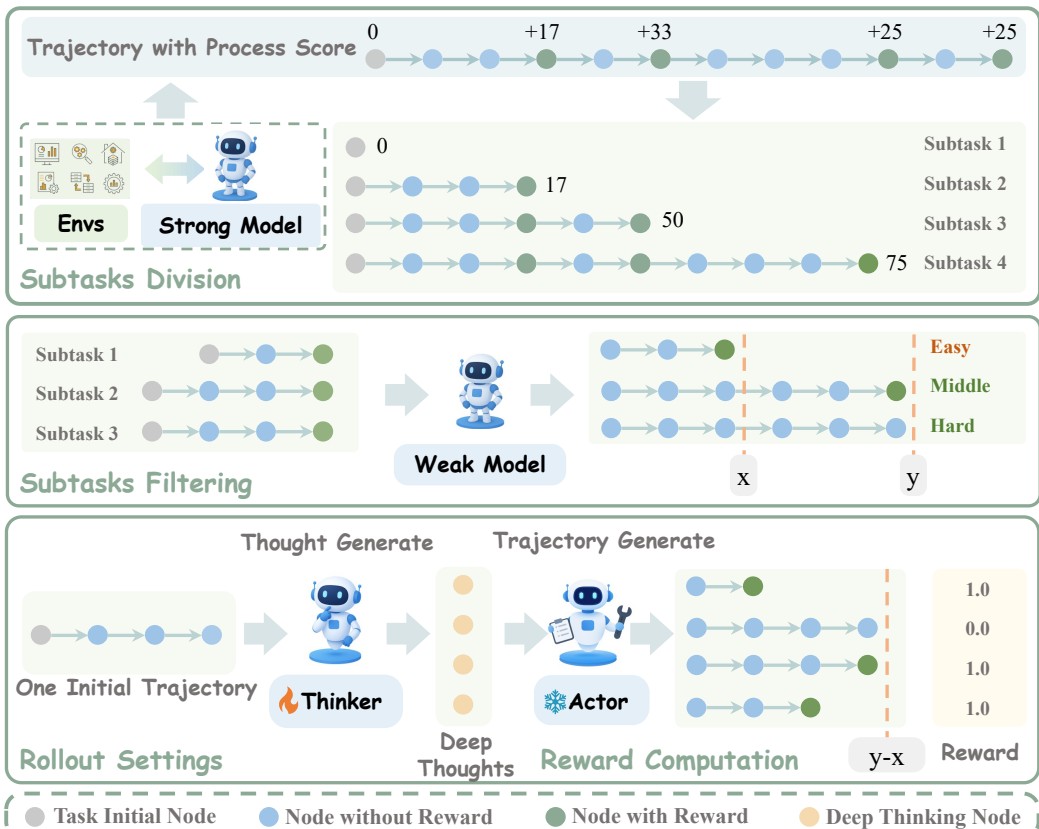

Figure 3: The Training pipeline of our professional thinker model.

actions. The output $d_i$ is then prepended to the actor's subsequent context, as shown in Eq. 3. In this way, the thinker directly influences the actor's behavior and helps prevent repeated mistakes.

$$a_{t+1} \sim \pi_\theta^{actor}(\cdot | u, o_o, a_1, o_1, \ldots, a_i, o_i, d_i, \ldots, a_j, o_j, d_j, \ldots, a_t, o_t). \qquad (3)$$

This architecture also enables flexible implementation. The thinker can be the same model as the actor operating under a different prompt, or a larger, more specialized model dedicated to complex reasoning. More details can be seen in Appendix B.1.

## 3.2 Environment Construction for Thinker Training

The main challenge to train the thinker role lies in **designing a stable and well-defined reward function**. To address this, we adopt an indirect evaluation strategy based on a more stable metric: the task completion performance of the actors. If task performance improves, we consider the thinker's thoughts beneficial; otherwise, we regard them as detrimental.

Another challenge lies in how to **allocate the reward signal to each sampled thought**. In a long trajectory, only a single reward signal is available, while multiple deep-thinking nodes may appear during the evaluation stage. Our experiments show that if all deep-thinking nodes within a trajectory share the same reward, RL training can easily become unstable or even collapse. A likely cause is that the training noise becomes excessively large when multiple nodes inherit the same feedback. To mitigate this, we retain only one deep thinking node per trajectory during training.

We construct a stable training environment to implement the above designs, as shown in Figure 3, which consists of four key components: ***sub-task division, sub-task filtering, rollout settings, and reward computation.*** More details of our training pipeline are shown in Appendix B.2.

**Sub-tasks Division**    To establish sufficiently meaningful and evaluable milestones as the starting points of sub-tasks, we divide the successful trajectories. For each task in our training dataset, we

are given a user instruction $u_0$ and an initial environment state $s_0$. Using a strong model $\pi^{strong}$, we generate an execution trajectory $traj^{strong}$ for the task. Based on this trajectory, we partition the task into multiple sub-tasks according to the process score, as defined in Eq. 4. Each sub-task inherits the initial environment state $s_0$ and the corresponding trajectory $traj_i^{strong}$ up to the sub-task's starting point.

$$(u_0, s_0) \rightarrow traj^{strong} \rightarrow [traj_1^{strong}, traj_2^{strong}, \ldots, traj_n^{strong}]. \tag{4}$$

**Sub-tasks Filtering**  We aim for the thinker model to concentrate on learning solutions for non-trivial and challenging tasks. Therefore, we classify all the sub-tasks into three levels of difficulty (easy, middle, and hard), and exclude those identified as "easy". To assess task difficulty, we employ a weak model to generate execution trajectories, which is shown in Eq. 5. First, we set two thresholds, $x$ and $y$, where $0 < x < y$. Then, we let the weak model generate $y$-step trajectories from the start state of each sub-task. A sub-task is assigned to the easy-level if the weak model completes it within $x$ steps, to the medium-level if completion requires between $x$ and $y$ steps, and to the hard-level otherwise.

$$(u_0, s_0, traj_i^{strong}) \rightarrow traj_i^{weak}. \tag{5}$$

**Rollout Settings**  We reuse the trajectory $traj_i^{weak}$ from the previous step to construct a context that may contain mistaken or unreasonable actions, thus making the intervention of the thinker role necessary. Before each sampling session, we initialize the environment to the current state of the sub-task by the initial state $s_0$ and two trajectories $traj_i^{strong}$ and $traj_i^{weak}$. The length of the trajectory $traj_i^{strong}$ is arbitrary (can be zero), and the length of the trajectory $traj_i^{weak}$ is fixed to $x$. Then, the trainable thinker model generates its deep thoughts $m$ times depending on the same historical trajectory, as shown in Eq. 6.

$$(u_0, s_0, traj_i^{strong}, traj_i^{weak}) \rightarrow [d_{i,1}, d_{i,2}, ...d_{i,m}]. \tag{6}$$

**Reward Computation**  Based on the different sampled deep thoughts, a frozen actor model is invoked to execute a maximum of $(y - x)$ steps, resulting in a total of $m$ trajectories $traj^{actor}$, as shown in Eq. 7.

$$(u_0, s_0, traj_i^{strong}, traj_i^{weak}, d_{i,j}) \rightarrow traj_{i,j}^{actor} \rightarrow r_{i,j}. \tag{7}$$

Finally, we will calculate the rewards by the task completion performance of the trajectories: If the process score in a trajectory $traj_{i,j}^{actor}$ improves, the reward $r_{i,j} = 1.0$; otherwise, the reward $r_{i,j} = 0.0$. By evaluating a deep thought with one trajectory and giving it a reward $r \in [0, 1]$, we simplify the complex reward assignment problem into a clear, one-to-one causal judgment.

## 4 EXPERIMENTS

In the experiments section, we first present our experimental setup, including the training and inference settings, datasets, evaluation metrics, and baseline methods. We then analyze the main results, which show that our approaches improve the performance of both base models and well-trained agents. Next, in Section 4.3, we examine whether our methods enhance the exploration behavior of agents. Section 4.4 provides an ablation study on the thinker role, highlighting the importance of training a specialized thinker model and the effectiveness of our training pipeline. Finally, in Section 4.5, we investigate the impact of deep thinking frequency on performance.

### 4.1 SETTINGS

**Training and Inference Settings**  Our main work is to train a professional model **Exp-Thinker** as the thinker role in our **TTExplore** framework. We choose the Qwen2.5-7B-Instruct (Hui et al., 2024) as the backbone, and use Supervised Fine-Tuning (SFT) and Reinforcement Learning (RL) to train it. For the SFT stage, we use the TRL (von Werra et al., 2020) training framework and train 3 epochs. For the RL stage, we use the Verl (Sheng et al., 2024) training framework and train several epochs until the model converges. We also train actor models for further experiments, which are based on Qwen2.5-7B-Instruct and LLaMA3-8B-Instruct (Dubey et al., 2024), respectively. More details can be seen in Appendix C.1.

Table 1: **Main Results of Our Methods.** The first column and second column list the *Actor* role and the *Thinker* role in our proposed **TTExplore** framework, respectively. If the *Thinker* role is *No*, the framework back to the **ReAct** method, where the actor produces short thoughts. **Exp-Thinker** is trained specifically for the thinker. The last column shows the average results across all tasks. Note that * indicates the task is seen during actor model training and treated as in-domain evaluation.

| Actor | Thinker | In-Domain | | Out-Of-Domain | | | Mean |
|---|---|---|---|---|---|---|---|
| | | Alfworld | Sciworld | BabyAI | Jericho | PDDL | |
| *Large Models* | | | | | | | |
| GPT-4o | No | 79.90 | 76.90 | 64.10 | 34.00 | 69.80 | 64.94 |
| Qwen2.5-72B | No | 93.84 | 76.74 | 70.07 | 30.87 | 70.50 | **68.40** |
| *Fine-Tuned Agents* | | | | | | | |
| Agent-FLAN | No | 79.70 | 10.90 | 35.30 | 10.10 | 25.50 | 32.30 |
| AgentGym | No | 76.90 | **47.30** | 61.40* | 12.90 | 16.60 | 43.02 |
| AgentGen | No | 47.60 | 13.90 | 39.40 | 10.80 | 36.40 | 29.62 |
| AgentRefine | No | 63.80 | 42.60 | 50.40 | **32.30** | **37.80** | 45.38 |
| ATLAS | No | **84.50** | 42.02 | 80.98* | 18.21 | 15.84 | **48.31** |
| *Our Methods* | | | | | | | |
| LLaMA3-8B | No | 32.15 | 20.31 | 42.44 | 14.23 | 29.94 | 27.81 |
| | Exp-Thinker | **63.49** | **55.43** | **58.15** | **26.31** | **30.06** | **46.69** |
| Qwen2.5-7B | No | 80.03 | 39.82 | 50.16 | 8.95 | **25.38** | 40.87 |
| | Exp-Thinker | **94.96** | **72.69** | **63.12** | **19.94** | 20.87 | **54.32** |
| LLaMA3-Actor | No | 78.92 | 54.87 | 35.55 | 15.98 | 20.19 | 41.10 |
| | Exp-Thinker | **87.75** | **63.14** | **46.41** | **17.11** | **32.11** | **49.30** |
| Qwen2.5-Actor | No | 97.76 | **83.07** | 50.62 | 17.44 | 29.76 | 55.73 |
| | Exp-Thinker | **98.50** | 82.27 | **60.25** | **17.64** | **31.43** | **58.02** |

All experiments are conducted using vLLM (Kwon et al., 2023) as the inference engine. The temperature is fixed at $0.0$, corresponding to greedy decoding. For all test tasks, we limit the maximum number of interaction steps to 50. We trigger the thinker at a fixed frequency, which means after every $n$ steps of interaction, the thinker is activated. In our experiments, we set $n = 6$.

**Datasets, Metrics and Baselines** We evaluate our methods on five text-based embodied tasks, including ALFworld (Shridhar et al., 2020), Sciworld (Wang et al., 2022), BabyAI (Chevalier-Boisvert et al., 2018), PDDL (Vallati et al., 2015), and Jericho (Hausknecht et al., 2020). We follow the metrics, few-shot examples, and action space descriptions in Agentboard (Chang et al., 2024). We apply **Process Score** as the metric, which is in the range $0.0$ to $100.0$. Specifically, we classified the Alfworld and Sciworld as in-domain tasks, and the other three as out-of-domain tasks. More details can be seen in Appendix C.2.

In our experiments, we use LLaMA3-8B-Instruct, Qwen2.5-7B-Instruct, and their continue fine-tuned agents (LLaMA3-Actor and Qwen2.5-Actor) as the *actor* role. Then, we use our specially trained model **Exp-Thinker**, which is based on Qwen2.5-7B-Instruct, as the *thinker* role. We compared our method with large models and fine-tuned agents: (1) **Large Models**, including GPT-4o (Hurst et al., 2024) and Qwen2.5-72B-Instruct (Hui et al., 2024); (2) **Fine-Tuned Agents**, including Agent-FLAN (Chen et al., 2024), AgentGym (Xi et al., 2024), AgentGen (Hu et al., 2024), AgentRefine (Fu et al., 2025), and ATLAS (Wu et al., 2024).

## 4.2 MAIN RESULTS

**(1) The thinker role can enhance the performance of base models:** Our method yields substantial improvements for base models across both in-domain and out-of-domain tasks. When employing LLaMA3-8B and Qwen2.5-7B as the actor, our approach consistently outperforms the baseline setting (without a thinker) on nearly all tasks, achieving average performance gains of 14-19 points. For

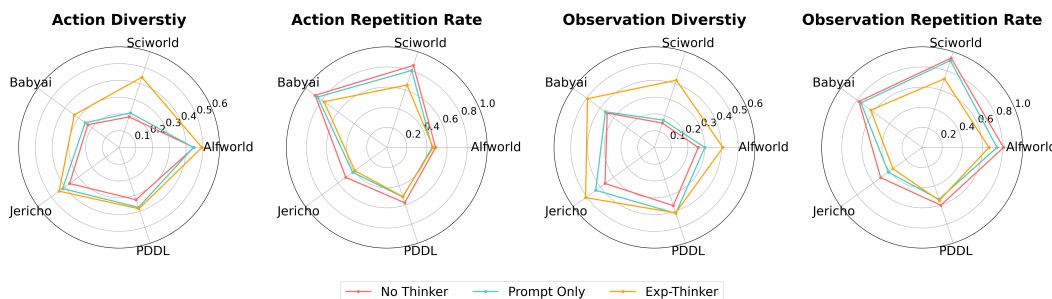

Figure 4: The quantitative analysis of exploration behavior for different methods.

instance, with LLaMA3-8B on in-domain tasks, the thinker model Exp-Thinker boosts performance from 32.15 and 20.31 to 63.49 and 55.43, which is an increase of more than 30 points. On out-of-domain tasks, the improvements remain notable, averaging nearly 10 points. These gains elevate the performance of the base model LLaMA3-8B to a level comparable with fine-tuned agents such as AgentRefine and ATLAS. Overall, these results highlight both the effectiveness and the strong generalization capabilities of our framework and the specially trained thinker.

**(2) The thinker role can enhance the performance of well-trained agents:** Prior studies have improved agent performance primarily through data construction for SFT or RL. Building on this, we further train LLaMA3-8B and Qwen2.5-7B on in-domain tasks, yielding stronger actor variants, LLaMA3-Actor and Qwen2.5-Actor. For LLaMA3-Actor, incorporating the thinker model improves performance across all tasks, with an average gain of 8 points. Qwen2.5-Actor, a well-trained agent on in-domain tasks, achieves the score of 97.76 and 83.07 on Alfworld and Sciworld, respectively. Even for such a strong agent, our method still enhances its out-of-domain performance, most notably on BabyAI, where accuracy increases from 50.62 to 60.25. This demonstrates that the traditional agent abilities improved by SFT and RL are orthogonal and complementary to the "test-time exploration" capability provided by our thinker. Our approach is not a replacement for existing training methods, but rather a powerful complement.

### 4.3 DOES THE THINKER IMPROVE THE ACTOR'S EXPLORATION BEHAVIOR?

This section provides a quantitative analysis of how our framework enhances the model's exploratory behavior. In complex environmental interaction tasks, agents often fail due to repetitive behaviors or overly localized exploration. To evaluate exploration more systematically, we introduce four metrics: *action diversity, action repetition rate, observation diversity, and observation repetition rate*. Given an execution trajectory $T(a_1, o_1, a_2, o_2, \ldots, a_n, o_n)$, **action diversity** is defined as the number of distinct actions divided by the trajectory length, while **action repetition rate** is defined as the proportion of top-$k$ most frequent actions within the trajectory. Similarly, **observation diversity** measures the ratio of distinct observations visited to the trajectory length, and **observation repetition rate** measures the proportion of the top-$k$ most frequently visited observations. In our experiments, we set $k = 3$. These four metrics allow us to quantify the exploratory behavior of a single trajectory. The higher diversity and lower repetition rate mean better exploratory behavior.

We compare the exploratory behaviors of LLaMA-8B-Instruct under different methods, as illustrated in Figure 4. The results indicate that incorporating the Exp-Thinker consistently increases both action and observation diversity while reducing repetition rates, demonstrating its effectiveness in enhancing exploration. These findings provide direct evidence for our central hypothesis: the thinker mitigates the "local exploration loops" and "repetitive trial-and-error" behaviors identified in the introduction. By encouraging more diverse actions and observations, the agent is steered away from unproductive cycles, leading to the performance gains reported in Table 1.

### 4.4 IMPACT OF DIFFERENT THINKING ROLES

We evaluate our **TTExplore** framework by incorporating different thinker roles. In our experiments, we adopt LLaMA3-8B-Instruct and Qwen2.5-7B-Instruct as the actor roles. For the thinker roles, we consider two categories: (1) **Base models**, including Qwen2.5-7B-Instruct, Qwen2.5-72B-Instruct,

Table 2: The performance of different thinker model training methods.

| Actor | Thinker | In-Domain | | Out-Of-Domain | | | Average |
| | | Alfworld | Sciworld | BabyAI | Jericho | PDDL | |
|---|---|---|---|---|---|---|---|
| LLaMA3-8B | No | 32.15 | 20.31 | 42.44 | 14.23 | 29.94 | 27.81 |
| | Qwen2.5-72B | 48.13 | 21.32 | 41.72 | 25.97 | **36.05** | 34.64 |
| | Qwen3-8B | **53.42** | **36.47** | 40.01 | **35.42** | 35.76 | **40.22** |
| | Qwen2.5-7B | 39.17 | 17.69 | **42.02** | 26.70 | 26.01 | 30.32 |
| | Distill Only | 47.13 | 33.15 | 38.88 | 23.19 | 28.58 | 34.19 |
| | RL Only | 48.00 | 35.59 | 44.33 | 21.43 | 25.54 | 34.98 |
| | Distill + RL | **63.49** | **55.43** | **58.15** | **26.31** | **30.06** | **46.69** |
| Qwen2.5-7B | No | 80.03 | 39.82 | 50.16 | 8.95 | 25.38 | 40.87 |
| | Qwen2.5-72B | **92.91** | **69.20** | **72.08** | **20.57** | 28.76 | **56.70** |
| | Qwen3-8B | 89.18 | 55.76 | 64.32 | 19.43 | **30.71** | 51.88 |
| | Qwen2.5-7B | 76.18 | 54.61 | 63.27 | 11.93 | 21.08 | 45.41 |
| | Distill Only | 90.79 | 70.82 | 64.41 | 18.18 | **21.33** | 53.11 |
| | RL Only | 83.27 | 66.70 | **69.55** | 10.59 | 21.12 | 50.25 |
| | Distill + RL | **94.96** | **72.69** | 63.12 | **19.94** | 20.87 | **54.32** |

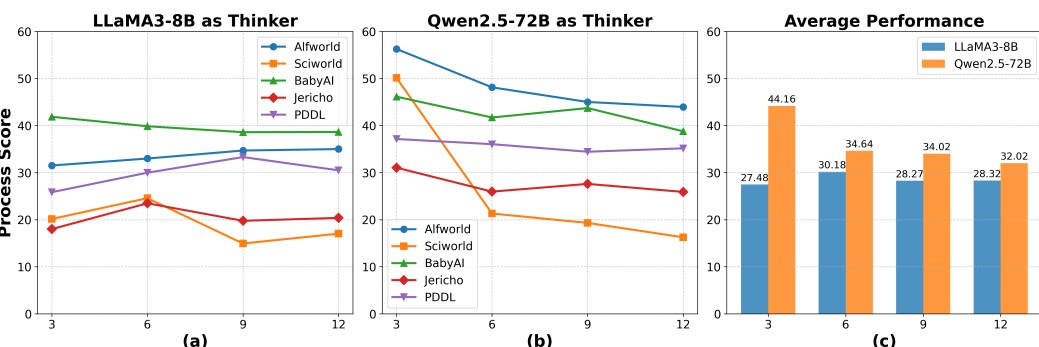

Figure 5: The performance of different deep thinking frequencies.

and Qwen3-8B (thinking mode). (2) **Trained thinker models**, obtained through different training strategies: *Distillation Only*, *Reinforcement Learning Only*, and *Distillation followed by RL*.

**Q1: Can we just use a large off-the-shelf model for the thinker role?** Results in Table 2 reveal a crucial insight: While larger model Qwen2.5-72B generally serve as a better thinker than its smaller counterparts, our specialized thinker (Distill + RL) consistently and substantially outperforms even the 72B model in most tasks. This demonstrates that targeted training on reasoning about environmental interactions is more effective than simply relying on the scale of a general-purpose model. It validates the necessity of our proposed training pipeline.

**Q2: Why is the "Distill + RL" training pipeline the best approach?** Our ablation confirms the complementary roles of distillation and reinforcement learning. "Distill Only" offers a strong initialization but is limited by the static dataset, while "RL Only" struggles with a weak starting state. The combined "Distill + RL" approach proves most effective, using distillation to overcome cold-start challenges and reinforcement learning to further enhance the thinker's reasoning abilities in an interactive environment, allowing it to discover strategies beyond the initial supervision.

### 4.5 IMPACT OF DEEP THINKING FREQUENCY

We further evaluate the impact of different deep thinking frequencies on task performance. In this experiment, we used LLaMA3-8B-Instruct as the actor, paired either with itself or with a larger model, Qwen2.5-72B-Instruct, as the thinker. The thinker was invoked at intervals of 3, 6, 9, and 12 steps. The results, presented in Figure 5, show that the optimal deep thinking interval varies across

| Method | Alfworld | | Sciworld | | BabyAI | | Jericho | | PDDL | | Mean | Total | Cost |
| | Score | Time | Score | Time | Score | Time | Score | Time | Score | Time | Score | Time | Ratio |
|---|---|---|---|---|---|---|---|---|---|---|---|---|---|
| ReAct | 78.9 | 71 | 33.4 | 95 | 49.3 | 91 | 7.3 | 53 | 23.1 | 79 | 38.4 | **389** | 1.0 |
| Reflexion | 85.6 | 341 | 56.9 | 436 | 75.6 | 469 | 18.0 | 291 | 38.3 | 301 | 54.9 | 1838 | 4.7 |
| Best-of-N | 96.3 | 357 | 70.5 | 443 | 84.2 | 467 | 26.7 | 295 | 54.0 | 362 | 66.3 | 1925 | 5.0 |
| TTExplore | 91.9 | 111 | 75.9 | 116 | 69.4 | 130 | 19.2 | 99 | 21.4 | 104 | 55.5 | 559 | 1.4 |
| TTExplore (N) | 99.4 | 589 | 91.1 | 606 | 85.3 | 687 | 31.5 | 507 | 43.8 | 519 | **70.2** | 2907 | 7.5 |

Table 3: Compare the time cost and performance with different methods. The TTExplore (N) variant refers to applying Best-of-N sampling on top of TTExplore. The score refers to the process score of the task, and the unit of time is seconds.

tasks. On average, when the thinker is a smaller model, deep thinking every 6 steps yields the best performance, whereas with a larger thinker model, deep thinking every 3 steps is most effective. We attribute this difference primarily to the quality of deep thinking. High-quality thinking can guide the agent toward more effective problem-solving and exploration, whereas low-quality thinking may interfere with the actor's normal execution. Increasing the frequency of thinking also introduces higher inference costs. Balancing performance and efficiency, we fixed the thinking interval at 6 steps in our experimental design. Nevertheless, this choice is not necessarily optimal across all tasks or model configurations. A promising direction for future work is to explore methods for dynamically adjusting the thinking frequency.

## 4.6 TIME COST OF DIFFERENT METHODS

Understanding the computational overhead introduced by our **TTExplore** method is essential for assessing its practical utility. Although our training process involves multiple sampling-and-evaluation iterations, our inference-time procedure uses only a single sample. Unlike traditional test-time scaling methods that rely heavily on multiple samples (e.g., Best-of-N or Reflexion), our method integrates deep thinking results directly into a single rollout. While the **Thinker** may consider multiple possible solution directions during deep thinking, the **Actor** performs actions sequentially, **without branching or backtracking**. This design is consistent with how humans explore real environments, where tasks are irreversible, and exploration must occur within a single action budget.

**Settings.** We compare our method TTExplore with standard ReAct and two representative test-time scaling methods: Reflexion and Best-of-N. Our method is orthogonal to Best-of-N, we additionally include a combined variant that applies Best-of-N sampling on top of TTExplore to illustrate their complementarity. In all experiments, we use Qwen2.5-7B-Instruct as the Actor and Exp-Thinker as the Thinker. Inference is executed using vLLM with parallel decoding and parallel environment stepping. The Actor's output is capped at 128 tokens, and the Thinker's at 512 tokens. For Reflexion and Best-of-N, we fix the number of samples to 5 to maintain a relatively fair computational comparison across all methods.

**Results.** As shown in Table 3, our TTExplore framework incurs a moderate computational overhead, running approximately $1.4\times$ slower than standard ReAct. This overhead is substantially lower than that of Reflexion ($4.7\times$) and Best-of-N sampling ($5.0\times$). Importantly, TTExplore remains fully compatible with Best-of-N sampling, enabling users to trade additional computational budget for further performance gains. When Best-of-N sampling is applied on top of TTExplore, the combined approach achieves the strongest overall performance.

## 5 RELATED WORKS

Agents based on Large Language Models (LLMs) have shown strong performance across various scenarios (Liu et al., 2023; Chang et al., 2024), including tool use, GUI navigation, and embodied tasks (Côté et al., 2018; Wang et al., 2022; Shridhar et al., 2020). Early approaches relied primarily on prompt-based methods (Yao et al., 2023; Chen et al., 2023) or framework design (Lin et al., 2023) to elicit effective behaviors. More recent studies focus on enhancing agent performance through two main directions: dedicated agent training and external knowledge injection.

Training-oriented approaches typically focus on collecting environments and tasks to strengthen agents' planning and reasoning abilities. Representative works include AgentTuning (Zeng et al., 2023), FireAct (Chen et al., 2023), Agent-FLAN (Chen et al., 2024), AgentOhana (Zhang et al., 2024), and AgentBANK (Song et al., 2024a). However, such agents often struggle with out-of-domain generalization. To address this, some studies expand the diversity of training environments and tasks. For instance, AgentGen (Hu et al., 2024) improves planning through synthesized tasks, while AgentRefine (Fu et al., 2025) enhances generalization by corrective data trajectories.

Alternatively, acquiring knowledge or experience about unknown environments has proven effective in boosting agent performance (Xiao et al., 2023; Yang et al., 2024a; Liu et al., 2025). ExpeL (Zhao et al., 2024) extracts rules from failed task trajectories to guide agents in new scenarios. Similarly, LWM (Xiang et al., 2023), KnowAgent (Zhu et al., 2024), and KWM (Qiao et al., 2024) utilize pre-acquired knowledge bases to support better planning. KnowSelf (Qiao et al., 2025) advances this idea by enabling agents to actively seek environment-specific knowledge in critical steps. Agent-RM (Xia et al., 2025) uses an external reward model to guide the agent generation with beam search.

In summary, existing methods primarily rely on two strategies: offline-memory, where agents are exposed to more environments during training, and offline-guidance, which depends on extensive pre-collected knowledge. Both approaches, however, lack test-time exploration, which means acquiring and utilizing environmental knowledge dynamically during interaction. We argue that agents should instead emulate human-like behavior: engaging in test-time exploration, continuously interacting with their environment to gather information, distill experiences, infer underlying rules, and ultimately accomplish tasks autonomously.

## 6 CONCLUSION

In response to the performance degradation of agents in interactive tasks on unknown environments, we propose the **TTExplore** framework. The framework introduces a thinker role that enables the actor role to conduct periodic deep thinking during extended interactions. By analyzing the interaction history, the thinker can clarify task objectives and progress, infer environmental implicit rules, and plan subsequent actions. These deep thoughts will regulate the actor's behavior, enhance its exploration diversity, and help it finish the task.

Furthermore, we provide a training pipeline to train the thinker models stably and train a professional thinker model **Exp-Thinker**. Evaluations on five text-based embodied tasks confirm the effectiveness and cross-domain generalization capability of our method.

In summary, we offer a robust solution to key limitations such as repeating actions, and local exploration traps. This work provides a feasible direction for enhancing the generalization of agents, especially the decision-making ability in unknown environments.

## ETHICS STATEMENT

We propose a framework and training pipeline that enable agents to better explore environmental knowledge, particularly implicit rules, during test time. All models and datasets used in this work are publicly available.

## REPRODUCIBILITY STATEMENT

Our prompts are provided in Appendix B.1, and the details of the training pipeline are described in Appendix B.2. The experimental settings are presented in Section 4.1, with training hyperparameters listed in Appendix C.1. The main code is included in the Supplementary Materials, and all code and models will be released upon acceptance of this work.

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

## A  THE USAGE OF LLMS

We use LLMs (e.g., ChatGPT and DeepSeek-V3.1) to help polish the paragraphs of our paper.

## B  METHOD DETAILS

### B.1  PROMPTS AND OUTPUT FORMATS

We show our prompts for the actor role and the thinker role in our framework, as shown in Figure B.1 and Figure 6. The output format the for actor role is formula $<think> \ldots </think><answer> \ldots </answer>$, which follows a ReAct-style (Yao et al., 2023). The output format of the thinker role is formula $<deepthink> \ldots </deepthink>$.

---

**Actor Prompt**

You are an Action Agent responsible for achieving a text-based task.
Now you need to finish a text-based task in an environment with multi-turn interaction.
Task Examples: [EXAMPLES]
Task Actions: [ACTION SPACE]
The Task: [TASK]
Initial Observation: [INIT OBSERVATION]
Attention:
1. You MUST provide your thought (one or two lines) before taking action.
2. You MUST issue only ONE action in each interaction stage.
Use the following format:

```
<think> put your thought here </think>
<answer> put your action here </answer>
```

Please provide your response to the task following the format strictly.

Figure 6: The prompt for the actor role.

---

**Thinker Prompt**

You are a Thinker Agent responsible for uncovering the implicit rules of the environment. You must analyze the history trajectory carefully and reason about any confusing feedback from the environment.
Here is the information about the task environment.
Task Actions: [ACTION SPACE]
The Task: [TASK]
Initial Observation: [INIT OBSERVATION]
History Trajectory: [INTERACTION HISTORY]
Attention:
1. If you think all the feedback in the history trajectory is reasonable, summarize the sub-goals you have completed and provide your next plan.
2. If you find the environment's feedback in the latest steps confusing, think carefully about possible reasons. Do not assume the environment is erroneous; instead, consider what hidden rules could explain the observations.
3. For any uncertainties, try to formulate hypotheses and design plans to verify them.
Use the following format for your response:

```
<deepthink> put your thought here </deepthink>
```

Figure 7: The prompt for the thinker role.

### B.2  THE TRAINING PIPELINE FOR THE THINKER MODEL

In this section, we present our approach for training a professional thinker model based on a small backbone. We train our thinker model in two stages: distillation learning and reinforcement learning.

Table 4: Training hyperparameters for Exp-Thinker.

| Parameter | SFT Stage | RL Stage |
|---|---|---|
| Learning rate | 1e-6 | 1e-6 |
| Batch size | 32 | 16 |
| Rollout number | – | 8 |
| LR scheduler | cosine | constant |
| Training epochs | 3 | 3 |
| BFloat16 | true | true |

Table 5: The number of tasks in different datasets.

| | Alfworld | Sciworld | BabyAI | Jericho | PDDL |
|---|---|---|---|---|---|
| Training tasks | 800 | 1,709 | – | – | – |
| Training sub-tasks | 1,093 | 5,139 | – | – | – |
| Evaluation tasks | 134 | 90 | 112 | 20 | 60 |

**Distillation Learning Stage**  When analyzing environmental interaction trajectories, large models demonstrate stronger deep reasoning abilities than smaller ones. To bridge this gap, we apply distillation learning, enabling the small model to **quickly acquire the reasoning patterns of the larger model**. As shown in the **left panel** of Figure 3, we assign a small model as the actor and a large model as the thinker, and use our TTExplore framework to generate task execution trajectories. Since the goal is to enhance the small model's reasoning ability, we only extract the thinker's outputs as training data.

**Reinforcement Learning Stage**  To further improve the thinker's performance, we train it with **reinforcement learning** using an indirect reward. We first decompose complex tasks into sub-tasks and filter them by difficulty. In the sub-tasks division stage, we use a fine-tuned agent, Qwen2.5-Actor, which is based on Qwen2.5-7B-Instruct as the strong model. In the sub-tasks filtering stage, we have two thresholds, $x$ and $y$, which are mentioned in Section 3.2. In our experiments, we set $x = 5$ and $y = 15$. We use a basic model, Qwen2.5-7B-Instruct, as the weak model. During the rollout stage, the trainable thinker generates multiple deep thoughts. In the reward computation stage, a fixed actor model executes subsequent trajectories based on the sampled thoughts, and the resulting task completion performance serves as the indirect reward signal. We update the thinker's parameters using the GRPO algorithm (Shao et al., 2024).

## C  EXPERIMENT DETAILS

### C.1  TRAINING DETAILS

**Training Details for Exp-Thinker**  We train a specialized thinker model, **Exp-Thinker**, using the method described in Section 3.2. The model is based on the Qwen2.5-7B-Instruct backbone. During the distillation learning stage (also referred to as the SFT stage), the model is fine-tuned on $10,797$ training samples. In the reinforcement learning stage, we use $6,232$ sub-tasks for training. Detailed hyperparameters for both stages are provided in Table 4.

**Training Details for LLaMA3-Actor and Qwen2.5-Actor**  We further fine-tune the LLaMA3-8B-Instruct and Qwen2.5-7B-Instruct models to obtain improved actor agents, denoted as **LLaMA3-Actor** and **Qwen2.5-Actor**, respectively. Both models are trained using reinforcement learning on $2,509$ tasks from the Alfworld and Sciworld environments. For LLaMA3-Actor, we incorporate additional SFT samples from AgentGym (Xi et al., 2024) to facilitate cold-start initialization. Note that we exclude the BabyAI task samples from the AgentGym dataset to ensure that BabyAI remains an out-of-domain evaluation environment.

### C.2  DATASETS DETAILS

We show the number of tasks in different datasets in Table 5. The training tasks are used for training actor models (LLaMA3-Actor and Qwen2.5-Actor), and the sub-tasks are used for training the thinker model (Exp-Thinker).

## C.3 THE EXAMPLES OF THE DEEP THINKING RESULTS.

In this section, we show some examples of the deep thinking results when using different models to play the role of thinker: our professional thinker model (Exp-Thinker), the Qwen2.5-72B-Instruct, and the Qwen3-8B (thinking mode).

---

**An Example of Deep Thinking Results from Our Exp-Thinker**

**Deep Thinking Result:**
The repeated failure to execute the command "go to outside" suggests that the environment does not recognize this command or that the location "outside" is not directly accessible from the workshop. This could be due to several reasons:
1. The command might need to be phrased differently.
2. There might be a specific action required to move to the outside, such as opening a door or using a different command.
3. The environment might require a more detailed command, such as specifying a door or a path to the outside.

Given the initial observation, the only door mentioned is to the hallway, which is open. This might be a clue that the hallway is the next logical step to reach the outside. However, the task specifies that the animals are in the "outside" location, and it's possible that the hallway might lead to the outside.

Next Plan:
1. Go to the door to the hallway.
2. Open the door if it is not already open.
3. Go to the hallway.
4. Look around to see if the outside is accessible from the hallway.
5. If the outside is accessible, proceed to find the animals and focus on the one with the longest life span, then the one with the shortest life span.

Action to take:
- go to the door to the hallway
- open the door to the hallway
- go to the hallway
- look around

---

Figure 8: An example of the deep thinking result produced by our professional thinker.

---

**An Example of Deep Thinking Results from Qwen2.5-72B-Instruct**

**Deep Thinking Result:**
The task is to put a clean plate in the countertop. The history trajectory shows that I have already taken a plate from cabinet 1 and placed it on countertop 1. However, the plate is not yet clean, which is a crucial part of the task. The sinkbasin 1 is likely the receptacle where I can clean the plate.

Subgoals completed:
1. Go to cabinet 1.
2. Take plate 1 from cabinet 1.
3. Go to countertop 1.
4. Put plate 1 in/on countertop 1.

Next plan:
1. Go to sinkbasin 1.
2. Clean plate 1 with sinkbasin 1.
3. Go to countertop 1.
4. Put the clean plate 1 in/on countertop 1.

---

> The feedback in the history trajectory is reasonable. The environment has provided clear and expected responses to the actions taken. There are no confusing elements in the feedback, and the actions align with the logical steps required to complete the task. The next step is to clean the plate, which I will do at the sinkbasin 1.

Figure 9: An example of the deep thinking result produced by Qwen2.5-72B-Instruct.

---

**An Example of Deep Thinking Results from Qwen3-8B**

Okay, let's see. The task is to put two soapbar in garbagecan. The initial observation shows various objects and receptacles. The history trajectory has the user opening cabinets 1, 2, and 3. Cabinet 1 has a cloth 1 and a soapbottle 1. Cabinet 2 has a candle 2 and toiletpaper 1. Cabinet 3 has a cloth 2.

Wait, the task is about soapbar, but in cabinet 1 there's a soapbottle. Maybe the soapbar is different from soapbottle? Or maybe the user needs to find soapbar somewhere else. The user hasn't found any soapbar yet. The cabinets 1-3 don't have soapbar. The other receptacles like countertop 1, sinkbasin 1 or 2, maybe those places could have soapbar? Or maybe the soapbar is in another cabinet, like cabinet 4? The user hasn't checked cabinet 4 yet.

The user's next step should be to check cabinet 4, since the initial observation mentions cabinet 4. Maybe the soapbar is there. Also, the garbagecan 1 is mentioned, so maybe after finding the soapbar, they need to take it and put it in the garbagecan. But first, need to find two soapbar.

Wait, the user has only opened cabinets 1-3. So the next logical step is to go to cabinet 4 and open it. If there's a soapbar there, then take it. Then check other receptacles like countertop 1, sinkbasins, etc. If the soapbar isn't in cabinet 4, maybe it's in another location. Also, the user might need to use the sinkbasin to clean something, but the task is about putting soapbar in garbage.

Another thought: maybe the soapbar is in the sinkbasin? Or maybe the soapbottle in cabinet 1 can be used to create soapbar? But the task says "soapbar", not "soapbottle". So perhaps the user needs to find two soapbar objects.

So the next action should be to go to cabinet 4 and open it. If that doesn't yield soapbar, then check other locations. Also, the user might need to check the countertop 1, as that's a common place for items.

Alternatively, maybe the soapbar is in the sinkbasin. But the user hasn't checked that yet. So the plan is to open cabinet 4 first. If that's empty, then check other receptacles.

So the current subgoals completed are opening cabinets 1-3. The next step is to go to cabinet 4 and open it. Then check if there's soapbar. If not, proceed to other locations.

But the user hasn't found any soapbar yet. So the next action is definitely to go to cabinet 4. If the environment's feedback is confusing, maybe there's a hidden rule about how objects are stored. For example, maybe soapbar is in a different cabinet or a specific receptacle. But based on the given info, the user hasn't checked cabinet 4 yet. So the next step is to go to cabinet 4 and open it.

**Deep Thinking Result:** The task requires placing two soapbar objects into the garbagecan. The history trajectory shows the user has opened cabinets 1-3, which contain cloth, soapbottle, candle, and toiletpaper, but no soapbar. The initial observation mentions a cabinet 4, which has not been explored yet. Since the task involves soapbar, and the only related object found is a soapbottle (not soapbar), the next logical step is to check cabinet 4. If soapbar is not there, other receptacles like countertop 1 or sinkbasins may need to be inspected. The environment's feedback is consistent so far, so no confusion is detected. The subgoal of opening cabinets 1-3 is complete. The next plan is to go to cabinet 4 and open it to search for soapbar.

Figure 10: An example of the deep thinking result produced by Qwen3-8B. We use its thinking mode, which means the model will give a long thinking in its default thinking mode first. After that, the model will give a deep thinking result depending on our requirements.

| Actor | Thinker | Alfworld | | Sciworld | | BabyAI | | Jericho | | PDDL | | Mean |
|---|---|---|---|---|---|---|---|---|---|---|---|---|
| | | Succ | Proc | Succ | Proc | Succ | Proc | Succ | Proc | Succ | Proc | Proc |
| LLaMA3-8B | No | 7.46 | 32.15 | 13.33 | 20.31 | 26.78 | 42.44 | 0.00 | 14.23 | 10.00 | 29.94 | 27.81 |
| | Exp-Thinker | 32.08 | 63.49 | 44.44 | 55.43 | 41.96 | 58.15 | 5.00 | 26.31 | 16.66 | 30.06 | **46.69** |
| Qwen2.5-7B | No | 68.65 | 80.03 | 33.33 | 39.82 | 37.50 | 50.16 | 0.00 | 8.95 | 18.33 | 25.38 | 40.87 |
| | Exp-Thinker | 90.29 | 94.96 | 65.55 | 72.69 | 50.00 | 63.12 | 5.00 | 19.94 | 8.33 | 20.87 | **54.32** |
| LLaMA3-Actor | No | 68.66 | 78.92 | 45.56 | 54.87 | 23.21 | 35.55 | 0.00 | 15.98 | 5.00 | 20.19 | 41.10 |
| | Exp-Thinker | 77.61 | 87.75 | 51.11 | 63.14 | 35.71 | 46.41 | 0.00 | 17.11 | 10.00 | 32.11 | **49.30** |
| Qwen2.5-Actor | No | 94.77 | 97.76 | 77.77 | 83.07 | 40.17 | 50.62 | 0.00 | 17.44 | 15.00 | 29.76 | 55.73 |
| | Exp-Thinker | 97.76 | 98.50 | 74.44 | 82.27 | 48.21 | 60.25 | 0.00 | 17.64 | 11.66 | 31.43 | **58.02** |

Table 6: Main results of our methods. The "Succ" refers to the success rate, and the "Proc" refers to the process score.

| **Actor** | **Thinker** | Alfworld | Sciworld | BabyAI | Jericho | PDDL | **Mean** |
|---|---|---|---|---|---|---|---|
| GPT-4o-mini | No | 47.63 | 86.23 | 67.23 | 20.72 | 44.65 | 53.29 |
| | GPT-4o-mini | 50.74 | 83.87 | 71.17 | 25.38 | 51.30 | 56.49 |
| | Qwen2.5-72B | 64.80 | 82.80 | 72.99 | 30.30 | 51.52 | **60.48** |
| LLaMA3-8B | No | 32.15 | 20.31 | 42.44 | 14.23 | 29.94 | 27.81 |
| | LLaMA3-8B | 33.00 | 24.58 | 39.86 | 23.49 | 29.99 | 30.18 |
| | Qwen2.5-72B | 48.13 | 21.32 | 41.72 | 25.97 | 36.05 | **34.64** |
| Qwen2.5-7B | No | 80.03 | 39.82 | 50.16 | 8.95 | 25.38 | 40.87 |
| | Qwen2.5-7B | 76.18 | 54.61 | 63.27 | 11.93 | 21.08 | 45.41 |
| | Qwen2.5-72B | 92.91 | 69.20 | 72.08 | 20.57 | 28.76 | **56.70** |
| Qwen2.5-14B | No | 84.32 | 66.80 | 66.57 | 27.76 | 51.62 | 59.41 |
| | Qwen2.5-14B | 85.88 | 76.51 | 71.81 | 29.63 | 54.76 | 63.72 |
| | Qwen2.5-72B | 88.93 | 75.38 | 72.38 | 29.42 | 57.69 | **64.76** |
| Qwen2.5-72B | No | 93.84 | 76.74 | 70.07 | 30.87 | 70.50 | 68.40 |
| | Qwen2.5-72B | 92.91 | 81.62 | 73.00 | 37.07 | 72.47 | **71.41** |

Table 7: The performance of our framework on different models.

## C.4 MORE DETAILS RESULTS OF OUR MAIN EXPERIMENTS.

Due to space limitations, we were unable to include task success rates in the Table 1. We add the success rates corresponding to the main experiments, as shown in Table 6.

## C.5 EVALUATE OUR FRAMEWORK ON MORE MODELS

We choose different models as the actor role and the thinker role for our framework. The results are shown in Table 7, which demonstrate the effectiveness of our framework.

## C.6 EVALUATE OUR METHOD WITH MORE BASELINES

To further validate the effectiveness of our approach, we compare TTExplore with a broader set of baselines. ETO (Song et al., 2024b) is a training method that leverages rejected trajectories via Direct Preference Optimization (DPO). Expel (Zhao et al., 2024), KnowAgent (Zhu et al., 2024), KWM (Qiao et al., 2024), and KnowSelf (Qiao et al., 2025) represent four knowledge-augmented approaches that rely on offline-collected environment knowledge to enhance agent performance. The comparative results are presented in Table 8.

Our experiments primarily use LLaMA3-8B and Qwen2.5-7B-Instruct as the backbone models, whereas most of the competing methods are implemented on LLaMA3.1-8B. As reflected by the ReAct baseline, LLaMA3.1-8B exhibits stronger inherent task-solving ability than LLaMA3-8B. Nonetheless, our method still achieves larger performance gains even without relying on offline-collected knowledge, outperforming most baselines.

| Backbone | Method | Know% | Actor Role | Thinker Role | Alfworld |
|---|---|---|---|---|---|
| Llama3.1-8B | ReAct | 0% | prompt | – | 27.61 |
| | Expel | 100% | prompt | – | 41.04 |
| | ETO | 0% | trained | – | 78.36 |
| | KnowAgent | 100% | trained | – | 75.37 |
| | WKM | 100% | trained | – | 77.61 |
| | Knowself | 100% | trained | – | **84.33** |
| Llama3-8B | ReAct | 0% | prompt | – | 7.46 |
| | TTExplore (ours) | 0% | prompt | trained | 32.08 |
| | ReAct | 0% | trained | – | 76.12 |
| | TTExplore (ours) | 0% | trained | trained | **80.60** |
| Qwen2.5-7B | ReAct | 0% | prompt | – | 76.10 |
| | TTExplore (ours) | 0% | prompt | trained | 91.90 |
| | ReAct | 0% | trained | – | 94.77 |
| | TTExplore (ours) | 0% | trained | trained | **97.76** |

Table 8: The performance of different methods on Alfworld benchmark. The last column is the **Success Rate** of the Alfworld.

| Actor | Thinker | Alfworld | Sciworld | BabyAI | Jericho | PDDL | **Mean** |
|---|---|---|---|---|---|---|---|
| Llama3-8B | No | 34.39 | 25.32 | 37.01 | 18.13 | 27.78 | 28.53 |
| | Thinker-v1 | 49.32 | 69.39 | **64.91** | **25.10** | **30.19** | **47.78** |
| | Thinker-v2 | **53.86** | **71.69** | 63.10 | 20.28 | 27.41 | 47.27 |
| Qwen2.5-7B | No | 78.92 | 33.38 | 49.29 | 7.29 | 23.09 | 38.39 |
| | Thinker-v1 | 88.62 | 68.52 | 64.46 | **17.74** | 20.92 | 52.05 |
| | Thinker-v2 | **91.04** | **78.20** | **66.25** | 16.38 | **22.60** | **54.89** |

Table 9: Compare the usage of binary rewards and more refined rewards in the training of the thinker.

### C.7 USING MORE REFINED REWARDS FOR THINKER TRAINING

In this section, we discuss whether using more refined rewards to replace the binary rewards, which used in our main experiment, will be better for the training of the thinker model. In our training method, the time (number of steps) required to achieve the first score improvement can serve as a finer-grained metric.

**Settings.** In this experiment, we introduced a step penalty of $-0.05$ for each generation step taken before the first score increase. The longer it takes for the actor to achieve improvement, the more cumulative penalty is applied. Using this reward, we trained a new model, **Thinker-v2**. For a fair comparison, we also retrained our original thinker under the original binary reward, 0 or 1, naming it **Thinker-v1**. Both models were trained on the same datasets, with identical hyperparameters, and for 400 steps, which the training curves show that both converge stably.

**Results.** We then evaluated Thinker-v1 and Thinker-v2 using two untrained models as actors, Qwen2.5-7B and LLaMA3-8B, within our TTExplore framework. Results indicate that denser rewards help more on in-domain data, while the effect is minimal on out-of-domain tasks.

### C.8 ADJUSTING THOUGHT NODES NUMBER PER TRAJECTORY FOR THINKER TRAINING

In this section, we investigate how the number of deep thought nodes within each trajectory affects the stability of RL training. In our initial configuration, the maximum trajectory length was set to 40 steps, and a deep thought trigger was applied every 6 steps, yielding approximately 6 deep thought outputs per trajectory. The trajectory-level reward—computed using the GRPO algorithm—was uniformly assigned to all deep thought nodes within that trajectory. Under this setup, training consistently collapsed after several dozen to roughly one hundred training iterations, exhibiting symptoms such as formatting degradation and increasingly garbled outputs. To mitigate this instability, our main experiment keep only a single deep thought node per trajectory, refer to as **Thinker-num1**.

| Actor | Thinker | Alfworld | Sciworld | BabyAI | Jericho | PDDL | Mean |
|-------|---------|----------|----------|--------|---------|------|------|
| Llama3-8B | No | 32.15 | 20.31 | 42.44 | 14.23 | 29.94 | 27.81 |
| | Thinker-num1 | 63.49 | 55.43 | 58.15 | 26.31 | 30.06 | **46.69** |
| | Thinker-num2 | 50.56 | 61.93 | 59.49 | 24.95 | 23.46 | 44.08 |
| | Thinker-num4 | 50.93 | 63.50 | 58.56 | 27.06 | 24.81 | 44.97 |
| Qwen2.5-7B | No | 80.03 | 39.82 | 50.16 | 8.95 | 25.38 | 40.87 |
| | Thinker-num1 | 94.96 | 72.69 | 63.12 | 19.94 | 20.87 | **54.32** |
| | Thinker-num2 | 90.11 | 74.53 | 63.68 | 19.15 | 22.28 | 53.95 |
| | Thinker-num4 | 91.04 | 65.85 | 63.56 | 21.23 | 23.28 | 52.99 |

Table 10: Compare the number of thought nodes per trajectory in the training of the thinker.

**Settings.** To further understand this phenomenon, we conducted two additional experiments by varying the number of deep-thought outputs per trajectory:

(1) We reduced the maximum trajectory length to 25 steps while maintaining a 6 step trigger interval, resulting in approximately four deep thought outputs per trajectory. Training under this setup did not collapse outright; however, validation accuracy rose initially and then declined, coinciding with a growing rate of formatting errors in sampled outputs. We refer to this model as **Thinker-num4**.

(2) We again set the maximum trajectory length to 25 steps but increased the trigger interval to 9 steps, producing roughly two deep thought nodes per trajectory. This configuration yielded stable training dynamics, clean output formatting, and validation accuracy that improved and then plateaued. We refer to this model as **Thinker-num2**.

**Results.** We compare **Thinker-num4** and **Thinker-num2** against our proposed single-thought-node training scheme. As shown in Table 10, the single-thought-node result (**Thinker-num1**) achieves the strongest overall performance, suggesting that excessive deep-thought density within a trajectory can destabilize policy optimization and degrade output reliability.

