# OpenReview forum: "Test-Time Exploration in Unknown Environments"
_ICLR.cc/2026/Conference — Submitted to ICLR 2026_

### Official Review · Reviewer_UZ3W · 2025-10-15

**Soundness:** 3
**Presentation:** 2
**Contribution:** 2
**Rating:** 6
**Confidence:** 4

**Summary:**

This paper introduces Test-Time Exploration (TTExplore), a framework designed to enhance large language model (LLM)-based agents operating in environments with implicit rules—that is, hidden constraints that cannot be directly observed and must be inferred through interaction. Existing LLM agents tend to fail in such environments, falling into repetitive trial-and-error loops. TTExplore addresses this by introducing a two-role architecture: an actor that performs ReAct-style actions, and a thinker that periodically conducts “deep thinking” to infer unobservable environmental rules from the interaction history and guide the actor’s future decisions.

To train the thinker, the authors develop a stable reinforcement learning pipeline that mitigates the challenges of sparse rewards. Evaluations on five text-based embodied benchmarks (ALFWorld, SciWorld, BabyAI, Jericho, and PDDL) show that TTExplore improves agent performance. The approach also enhances well-trained agents on out-of-domain tasks, demonstrating strong generalization and complementary benefits to traditional SFT/RL methods.

Overall, this work contributes a new perspective on test-time reasoning and exploration in LLM agents, providing both a conceptual framework and a training methodology for reasoning about implicit environmental rules.

**Strengths:**

- The paper presents an original and timely contribution to LLM-based agent research by focusing on test-time exploration—a crucial problem of reasoning under implicit environmental rules. The proposed thinker–actor framework introduces a conceptually clear and modular design that separates high-level reasoning from low-level execution, which is broadly applicable.

- From a technical quality perspective, the paper demonstrates solid methodological rigor through a carefully constructed reinforcement learning pipeline that tackles the instability of sparse rewards via task decomposition and difficulty filtering. The experiments are comprehensive, covering multiple embodied benchmarks with consistent and significant improvements over strong baselines.

- The paper is well written and structured, with clear motivation, diagrams, and examples that effectively illustrate the challenges of implicit rule reasoning.

- The work has good amount of contributions: it advances agent autonomy by enabling reasoning at test time, offering a complementary direction to existing fine-tuning or prompt-based methods. This framework could inspire further research on adaptive, self-reflective LLM agents operating in complex and uncertain environments.

**Weaknesses:**

- **Limited theoretical grounding:** The framework lacks a formal analysis of why the thinker–actor interaction improves exploration or generalization. A more principled explanation—e.g., through information-theoretic or reinforcement-learning formulations—would strengthen the conceptual foundation beyond empirical observation.

- **Evaluation scope and metrics:** The experiments focus primarily on text-based embodied environments (e.g., ALFWorld, SciWorld), which may not generalize to other modalities such as visual or tool-use agents. Furthermore, the evaluation metric (process score) captures task progress but not reasoning quality or exploration efficiency. Including qualitative analysis or human evaluation of “deep thinking” outputs would provide stronger evidence of reasoning improvement.

- **Ablation and comparison limitations:** Although the paper compares against ReAct and reflection-style methods, it omits other recent test-time reasoning approaches such as Reflexion (Shinn et al., 2023) or SwiftSage (Lin et al., 2023) in a direct, controlled setting. A more explicit comparison with those would clarify novelty and performance differences. As well, missing the comparison of more recent methods.

- **Scalability and efficiency concerns:** The framework introduces additional inference cost by invoking the thinker periodically. While the paper briefly discusses this, there is no quantitative analysis of compute trade-offs or latency, which is crucial for assessing practicality.

**Questions:**

- Clarification on the thinker–actor interaction mechanism: Could the authors elaborate on how the thinker’s “deep thoughts” are integrated into the actor’s context at inference time? Is there a gating or weighting mechanism for when to trust the thinker’s guidance, or are all thoughts appended uniformly?

- Reward assignment in thinker training: The paper mentions using a single deep-thinking node per trajectory to stabilize RL training. How sensitive is performance to this design choice? Would multiple thought nodes with appropriately discounted rewards destabilize training, or could a credit assignment scheme (e.g., temporal difference learning) be applied?

- Comparison with existing self-reflective frameworks: The paper cites ReAct and Reflexion but does not present a direct quantitative comparison under matched setups. How does TTExplore differ empirically or conceptually from Reflexion’s “verbal reinforcement” and from SwiftSage’s fast–slow thinking architecture?

- Generalization across modalities and domains: Since all experiments are on text-based embodied tasks, can the framework extend to multimodal or real-world robotic environments? Would the thinker’s reasoning process remain effective when observations are visual or high-dimensional?

- Ablation on the training pipeline: The “task decomposition” and “difficulty filtering” components are central to stability. Can the authors provide ablations showing how each contributes quantitatively to final performance?

---

> ### Author Response · Authors · 2025-11-21
>
> Dear reviewer,
> Thank you very much for your comments and professional advice. We would like to show the detailed responses as follows:
>
> ---
>
> ### Weaknesses:
>
> **Q1: Limited theoretical grounding.**
>
> Thank you for the insightful comment. We agree that providing a fully rigorous theoretical proof is challenging, but we offer an explanation grounded in commonly accepted RL formulations. The interaction between the actor and environment can be viewed within a partially observable Markov decision process (POMDP). In such settings, when an agent receives unexpected or low-information feedback from the environment, it is prone to falling into local exploratory loops or producing repetitive actions. This behavior is essentially a form of context imitation: the model overfits to similar local histories and collapses to low-diversity action patterns, ultimately exhausting the allowed step budget and failing the task.
>
> Our framework mitigates this issue by injecting the thinker's deep thoughts into the actor’s context. These deep-thinking tokens break the repetitive context patterns and introduce structured hypotheses, alternative plans, and uncertainty reasoning. This perturbation encourages the model to consider a broader range of action candidates, improving both action diversity and the diversity of states the agent visits. As shown in Section 4.3, our empirical results provide strong evidence for this effect: the thinker–actor interaction substantially increases action diversity as well as observation diversity, both of which are closely associated with improved exploration behavior.
>
> **Q2: Evaluation scope and metrics.**
>
> **(1) Generalization across modalities and domains.**
>
> We believe that the modality itself does not affect the core idea of our approach—namely, that an agent system should actively explore deep environmental information (such as implicit rules) during task execution. For multimodal or real-world tasks, the primary change is simply replacing textual observations with other modalities (e.g., images, audio, webpage screenshots). This substitution does not alter the Test-Time Exploration framework or the training procedure of the Thinker, although it may introduce some implementation-level challenges.
>
> In purely textual settings, the available observations are limited, enabling the Thinker to access and analyze the full interaction history. In visual tasks, however, each observation may consist of one or more images. Passing the entire visual history to the Thinker each time would likely exceed context limits. On the other hand, using a truncated history may result in incomplete information for the Thinker, reducing its ability to discover latent environmental rules.
>
> Therefore, we expect that applying our method to visual or multimodal tasks will require combining it with effective context-management strategies—such as summarization, visual feature compression, or image down-sampling—to maintain both efficiency and reasoning quality.
>
> In the future, we plan to explore more complex environments, such as GUI agent settings involving newly apps or functionalities, where exploration is more critical.
>
> **(2) Qualitative analysis or human evaluation of “deep thinking”.**
>
> In Section 4.3, we provide a brief analysis of how introducing deep-thinking nodes improves the agent’s exploratory behavior, such as increasing action diversity and enriching environmental observations. We also include several examples of deep-thinking outputs in Appendix C.3.
>
> Conducting a thorough qualitative analysis of deep thinking is indeed challenging, primarily for two reasons:
>
> **a.** It requires substantial human effort to determine whether the hypotheses and reasoning within the deep-thinking process accurately capture the true underlying rules of the environment.
>
> **b.** Even when the deep-thinking module correctly infers those rules, it is almost impossible to manually assess whether its expressed reasoning is actually helpful for the actor in correcting previous mistakes.
>
> For these reasons, evaluating deep thinking indirectly through the actor’s task performance becomes a more stable and efficient way to judge its effectiveness.

---

> > ### Author Response · Authors · 2025-11-21
> >
> > **Q3: Ablation and comparison limitations.**
> >
> > **(1) Difference from Reflexion.**
> >
> > Our method focuses on improving an agent system’s real-time environment-exploration capability. This differs substantially from Reflexion, whose core mechanism is to perform reflection only after a task has clearly failed. In contrast, our approach aims to explore environmental rules during task execution. We are motivated by two practical considerations:
> >
> > **a.** In many real-world scenarios, the environment cannot be reset arbitrarily, and repeated re-initialization and execution is infeasible.
> >
> > **b.** Even when the environment can be reset, complex tasks typically require multi-step interactions. Re-starting from scratch after each failure is prohibitively expensive.
> >
> > Our design therefore enables agents to explore the environment within a single execution trajectory, incorporating deep reasoning in real time to adjust their strategies more efficiently.
> >
> > **(2) Difference from SwiftSage.**
> >
> > Our method enhances the exploration capability of a two-role agent system, where the Actor and the Thinker have clearly defined functional responsibilities. The Actor handles direct observations (shallow information) and selects actions, while the Thinker analyzes the interaction history to infer hidden environmental rules (deep information). The two roles do not imply a strong–weak hierarchy; they may share the same underlying model or be instantiated by different models.
> >
> > SwiftSage, however, is built on a fundamentally different assumption: a strong model provides high-level planning, and a weak model grounds these plans into executable actions. This framework relies on the strong model having sufficient prior knowledge of the task and environment such that its plans are reliable. When the environment contains unknown or atypical rules that invalidate the strong model’s assumptions, SwiftSage may fail. In contrast, our method is designed for the Thinker to continuously interpret the environment and adapt to such unknowns in real time.
> >
> > **(3) Empirical comparison with related methods.**
> > We include comparisons with Reflexion and Best-of-N (with both reflection count N and sampling count N set to 5). The results are provided in response to Q4.
> >
> > **Q4: Scalability and efficiency concerns.**
> >
> > Our inference setup uses vLLM with parallel decoding across samples and parallel environment execution. In our experiments, the Actor’s max output length is 128 tokens, and the Thinker’s output is capped at 512 tokens.
> >
> > Taking Qwen2.5-7B-Instruct and our Exp-Thinker as an example, we measured the execution time of each method across tasks. On average, our TTE framework takes about 1.4 times longer to execute than the standard ReAct. This increase is expected due to additional deep-thinking steps. At the same time, our method remains compatible with Best-of-N, allowing users to trade extra compute for higher task success rates.
> >
> > | Method           | Alfworld Score | Alfworld Time(s) | Sciworld Score | Sciworld Time(s) | BabyAI Score | BabyAI Time(s) | Jericho Score | Jericho Time(s) | PDDL Score | PDDL Time(s) | Mean Score | Total Time (s) | Cost |
> > |-----------------|----------------|-----------------|----------------|-----------------|--------------|----------------|---------------|----------------|------------|--------------|------------|----------------|------|
> > | ReAct           | 78.9           | 71              | 33.4           | 95              | 49.3         | 91             | 7.3           | 53             | 23.1       | 79           | 38.4       | 389            | 1.0  |
> > | Reflexion (N times)      | 85.6           | 341             | 56.9           | 436             | 75.6         | 469            | 18.0          | 291            | 38.3       | 301          | 54.9       | 1838           | 4.7  |
> > | Best-of-N       | 96.3           | 357             | 70.5           | 443             | 84.2         | 467            | 26.7          | 295            | 54.0       | 362          | 66.3       | 1925           | 5.0  |
> > | TTE             | 91.9           | 111             | 75.9           | 116             | 69.4         | 130            | 19.2          | 99             | 21.4       | 104          | 55.5       | 559            | 1.4  |
> > | TTE+Best-of-N   | 99.4           | 589             | 91.1           | 606             | 85.3         | 687            | 31.5          | 507            | 43.8       | 519          | 70.2       | 2907           | 7.5  |

---

> > > ### Author Response · Authors · 2025-11-21
> > >
> > > ### Questions:
> > >
> > > **Q1: Clarification on the thinker–actor interaction mechanism.**
> > >
> > > We directly inject the Thinker’s deep-reasoning output into the Actor’s execution trajectory, following Eq. 3 in line 194. Concretely, we use a standard message-templating scheme to manage the Actor’s context, consisting of three roles: system, user, and assistant. In this framework, environmental feedback is placed in the user content, while the Actor’s short thoughts and actions, together with the Thinker’s deep-reasoning outputs, are placed in the assistant content.
> > >
> > > **Q2: Reward assignment in thinker training.**
> > >
> > > **(1) Adjusting the number of thought nodes per trajectory.**
> > >
> > > In our initial setup, the maximum trajectory length during training was limited to 40 steps, and a deep-thought trigger was applied every 6 steps, resulting in an average of 6 deep-thought outputs per trajectory. The overall trajectory-level evaluation—computed using the GRPO algorithm—was evenly distributed across all deep-thought outputs within that trajectory. Under this configuration, training collapsed after several dozen to around one hundred training steps, exhibiting symptoms such as formatting corruption and garbled outputs.
> > >
> > > Following your suggestion, we conducted two additional experiments by adjusting the number of deep-thought steps per trajectory:
> > >
> > > **a. Experiment 1:** We reduced the maximum trajectory length to 25 steps while keeping the deep-thought trigger interval at every 6 steps, yielding an average of 4 deep-thought outputs per trajectory. Under this setup, training did not collapse, but the validation accuracy rose initially and then declined. This turning point coincided with an increasing number of formatting errors in sampled outputs. We refer to the resulting model as **thinker-num4**.
> > >
> > > **b. Experiment 2:** We again limited the trajectory length to 25 steps but increased the trigger interval to every 9 steps, resulting in an average of 2 deep-thought outputs per trajectory. This configuration produced stable training dynamics, with clean output formats and validation accuracy that rose and then plateaued. We refer to this model as **thinker-num2**.
> > >
> > > We then compared these two models with our proposed single deep-thinking node per trajectory conditional training regime. The results indicate that the single-thought-per-trajectory setup yields the best performance.
> > >
> > > | Actor        | Thinker       | Alfworld | Sciworld | BabyAI | Jericho | PDDL  | Mean  |
> > > |--------------|---------------|----------|----------|--------|---------|-------|--------|
> > > | Llama3-8B    | No            | 32.15    | 20.31    | 42.44  | 14.23   | 29.94 | 27.81 |
> > > |              | Thinker-num1  | 63.49    | 55.43    | 58.15  | 26.31   | 30.06 | **46.69** |
> > > |              | Thinker-num2  | 50.56    | 61.93    | 59.49  | 24.95   | 23.46 | 44.08 |
> > > |              | Thinker-num4  | 50.93    | 63.50    | 58.56  | 27.06   | 24.81 | 44.97 |
> > > | Qwen2.5-7B   | No            | 80.03    | 39.82    | 50.16  | 8.95    | 25.38 | 40.87 |
> > > |              | Thinker-num1  | 94.96    | 72.69    | 63.12  | 19.94   | 20.87 | **54.32** |
> > > |              | Thinker-num2  | 90.11    | 74.53    | 63.68  | 19.15   | 22.28 | 53.95 |
> > > |              | Thinker-num4  | 91.04    | 65.85    | 63.56  | 21.23   | 23.28 | 52.99 |
> > >
> > >
> > > **(2) Adjusting the reward assignment scheme.**
> > >
> > > We believe that modifying the reward assignment strategy—such as redistributing rewards more selectively or introducing discount factors—does not fundamentally resolve the training challenges associated with having multiple deep-thought nodes within a single trajectory. This is because it is inherently difficult to determine whether earlier or later deep-thought nodes contribute more to the final outcome; within a trajectory, each deep-thought node may have an equally uncertain chance of being helpful or ineffective.
> > >
> > > **Q3: Comparison with existing self-reflective frameworks.**
> > >
> > > See Weaknesses Q3 and Q4.
> > >
> > > **Q4: Generalization across modalities and domains.**
> > >
> > > See Weaknesses Q2 (1).

---

> > > > ### Author Response · Authors · 2025-11-21
> > > >
> > > > **Q5: Ablation on the training pipeline.**
> > > >
> > > > **(1) Task Decomposition.**
> > > >
> > > > Without task decomposition, the agent’s required number of steps to finish a task becomes uncontrollable, meaning we cannot constrain the trajectory length. For instance, certain SciWorld tasks require dozens of interaction steps to complete. We compared our task-decomposed single-thought-per-trajectory setup with the no-decomposition setting, where each trajectory contains multiple deep-thought outputs (2 or 4 per trajectory). The corresponding results are provided in Questions Q2(1).
> > > >
> > > > **(2) Difficulty Filtering.**
> > > >
> > > > The main purpose of difficulty filtering is to remove short-horizon subtasks. Our training procedure requires that, before sampling a deep-thought output, the trajectory must already contain a sufficient number of environment interactions. This is essential for enabling the Thinker to analyze meaningful patterns from the interaction history. Extremely simple subtasks usually finish within just a few steps and therefore cannot satisfy this requirement. As a result, we are unable to perform an ablation that removes difficulty filtering, since doing so would violate the prerequisites of our training setup.

---

> > > > > ### Author Response · Authors · 2025-11-26
> > > > >
> > > > > Dear Reviewer UZ3W,
> > > > >
> > > > > We sincerely thank you for your constructive comments and valuable suggestions! We have discussed these issues and look forward to your feedback. If you have any further questions, we would be happy to continue the discussion.
> > > > >
> > > > > Thank you again!
> > > > >
> > > > > Authors

---

### Official Review · Reviewer_aoxx · 2025-10-21

**Soundness:** 2
**Presentation:** 3
**Contribution:** 1
**Rating:** 2
**Confidence:** 4

**Summary:**

This paper attempts to improve agent performance on partial-observable tasks with implict rules. The authors propose the framework **Test-Time Exploration**, which introduces another agent as *thinker* in additional to the actor, and trains a corresponding *thinker* model **Exp-Thinker**. Training this model requires trajectory sampled from a stronger model. The framework utilizes these trajectories through the following steps: it first decomposes a trajectory into sub-tasks (**Sub-Task Division**). Then, the system filtered out those sub-tasks that the actor agent has already been able to solve (**Sub-tasks Filtering**);  Then, the trainable thinker model generates thoughts by comparing strong trajectory that is generated by a stronger model with trajectory of current agent. Eventually, the **Reward Computation** step gives binary rewards: when the thoughts generated help improve the further actions, it will be rewarded with 1, otherwise 0. The paper compares their method with large models e.g. GPT-4o and fine-tuned models for agent tasks, and observe improvement in domains including ALFWorld, SciWorld, BabyAI, Jericho and PDDL.

**Strengths:**

- The paper is well-written and easy to follow.
- The paper discusses a way of fine-tuning a model using SFT&RL for usage as an agent, which could be of interest of a part of the research community.

**Weaknesses:**

Overall I have three major concerns regarding the idea and implementation of the paper. Specifically
1. Lack of novelty.  The paper can be regarded as distilling the knowledge gotten from the environment to teach an action model. This idea has long been researched by a sequence of papers: including
 - [O3D: Offline Data-driven Discovery and Distillation for Sequential Decision-Making with Large Language Models](https://arxiv.org/abs/2310.14403): Offline framework that discovers reusable skills from trajectories and distills them into an LLM agent. it tests on **ALFWorld** and WebShop(which is more complex then ALFWorld).
- [Embodied Multi-Modal Agent trained by an LLM from a Parallel TextWorld](https://openaccess.thecvf.com/content/CVPR2024/papers/Yang_Embodied_Multi-Modal_Agent_trained_by_an_LLM_from_a_Parallel_CVPR_2024_paper.pdf): Trains a VLM in the visual ALFWorld by distilling an LLM expert’s reflections from the parallel text world;
- [Structured Agent Distillation for Large Language Model](https://arxiv.org/abs/2505.13820): Distills a ReAct-style teacher by segmenting trajectories into [REASON] and [ACT] spans with segment-specific losses. Benchmarks include **ALFWorld**, HotPotQA-ReAct, WebShop.

**These methods share similar philosophy and tested on similar benchmarks. But none of them is included in related works or tested as baseline.**

2. I doubt if "test time exploration"  is a suitable name for what is actually implemented in the paper. While the research community has observed advances in test time scaling methods in domains such as large reasoning model, this improvement/novelty may not naturally transfer to agents. This is because, under partial observable environments, it is natural, and actually a **must**, for agents to do "test time exploration".  In fact, planning and exploration has long been a core research topic for agents.  This actually has been reflected in the related work section of this paper: "KnowAgent (Zhu et al.,2024), and KWM (Qiao et al., 2024) utilize pre-acquired knowledge bases to support better planning. KnowSelf (Qiao et al., 2025) advances this idea by enabling agents to actively seek environment-specific knowledge in critical steps. " Moreover, I don't think the method really fits the name "test time exploration". The core idea of this paper is to train another model to guide the action agent, and I don't think it has much to do with "test time exploration".

3. The critical weakness of the method I notice, is that it requires **presence of ground truth trajectory**. To enable any of the proposed steps, including sub-task division and filtering, as well as generating the thoughts, it requires the presence of ground truth. In current pipeline, such ground truth trajectories are sampled from strong models. This is doable for simple environments, such as ALFWorld, but this may not hold true for more complex environments, such as WebShop(which is tested in previous ) and WebArena. The authors need to show that their method also work for more complex environments. There are at least two ways to do it: one is to show that their thinker model can directly generalize to more difficult domains, and the second one is to show that such trajectory can still be collected in more complex domains AND the framework can still utilize them well.

**Questions:**

Following the weaknesses described above, the major questions are as follows:
- How do you think the method is different from the previous works? Could you demonstrate that your methods are better than the previous methods through experiments and discussions(insights)? We may need to see results including
     - baseline methods on ALFWorld and WebShop
     - and your results on WebShop(and maybe other more complex domains)
- Why "test time exploration" is a suitable name for what is actually implemented in the paper?
- Can the thinker model work on more complex domain? This links to question 1 and weakness point 3.

Minor points include:
- Why adding thoughts at fixed steps is a good alternative for adding thoughts when it is needed? The later one seems much more intuitive. Authors are encouraged to present further experiment results and justify their choice.

---

> ### Author Response · Authors · 2025-11-21
>
> Dear reviewer,
> Thank you very much for your comments and professional advice. We would like to show the detailed responses as follows:
>
> ---
>
> **Q1：Lack of novelty; the proposed method is viewed as knowledge distillation to teach an action model.**
>
> Our method is fundamentally different from knowledge distillation:
>
> (1) **Our core insight is that an agent often fails in unknown environments not because it lacks general commonsense knowledge**, but because it does not understand the implicit rules of the environment. We aim for an agent system that, like humans, can infer such latent rules on the fly through observation, environmental feedback, and reasoning—thereby enabling more reliable task completion.
>
> Prior works have indeed recognized the importance of task- or environment-specific knowledge and implicit rules (e.g., O3D, *Structured Agent Distillation for Large Language Models, KnowAgent*). However, these approaches extract such knowledge offline, then inject the resulting insights, examples, or distilled rules into the actor agent’s context.
>
> A central distinguishing aspect of our work is the emphasis on test-time autonomous exploration—the ability to discover implicit knowledge or rules during inference, driven by real-time environmental feedback, rather than relying on offline exploratory trajectories. Human exploration provides an analogy: we do not rely solely on prior knowledge, nor do we only reflect after failures (such as Reflexion). In new environments—even when faced with unknown rules—we can often succeed on the first attempt. This process involves interpreting surface-level feedback (short reasoning, as in ReAct) and also pausing to hypothesize and deeply analyze latent patterns emerging across multiple interaction steps (deep reasoning and exploration).
>
> In many agent settings, it is infeasible to gather extensive offline trajectories (due to time or environment cost), or to repeatedly reset the environment to the same initial state. This creates the need for human-like, real-time exploratory reasoning at test time—which is exactly the core motivation of our work.
> Based on this perspective, our contributions are twofold:
>
> a. We propose the Test-Time Exploration framework, in which the “thinker” role engages in deep thinking to help the actor uncover implicit environmental rules. Our framework does not rely on model training; the actor and thinker can be the same model, or a stronger model may serve as the thinker. Table 5 in the appendix shows the effectiveness of the framework even without additional training. We evaluate both closed-source and open-source models of varying sizes, and all achieve consistent improvements over the ReAct baseline.
>
> b. We introduce a training method for the thinker, enabling small-scale models to perform more effective deep thinking. The results are shown in Table 1 of the main paper. The goal of training is to help small models learn to infer exploratory patterns from one test-time environment trajectory—not to memorize environment-specific rules—which is why we further test on out-of-distribution tasks.
> (2) Thank you for the three paper links. After reviewing them, we believe they fall under offline environment knowledge extraction, and we plan to include them in the Related Work section. We also compare their results on ALFWorld with ours as follows:
>
> | Backbone         | Method                   | Alfworld (success rate) |
> |------------------|---------------------------|---------------------------|
> | GPT-4            | O3D                       | 91.00                     |
> | GPT-3.5          | O3D                       | 41.00                     |
> | multi-agents     | EMMA *                    | 82.00                     |
> | Llama-7B         | Structured Agent Distillation | 68.00                 |
> | Llama3.1-8B      | ReAct                     | 27.61                     |
> |                  | Reflexion                 | 51.49                     |
> |                  | Expel                     | 41.04                     |
> |                  | ETO                       | 78.36                     |
> |                  | KnowAgent                 | 75.37     |
> |                  | WKM                       | 77.61       |
> | **Ours**         |                           |                           |
> | Llama3-8B        | ReAct                     | 7.46          |
> |                  | TTExplore (ours)          | 32.08                     |
> | Llama3-Actor     | ReAct                     | 76.12     |
> |                  | TTExplore (ours)          | 80.60                     |
> | Qwen2.5-7B       | ReAct                     | 76.10        |
> |                  | TTExplore (ours)          | 91.90   |
> | Qwen2.5-Actor    | ReAct                     | 94.77                     |
> |                  | TTExplore (ours)          | **97.76**        |

---

> > ### Author Response · Authors · 2025-11-21
> >
> > **Q2: Whether “test-time exploration’’ accurately describes what the paper implements.**
> >
> > Thank you for your thoughtful comments and suggestions. However, we respectfully disagree with part of your interpretation. Below we clarify our perspective:
> >
> > **(1) Exploration is not spontaneous for an agent.**
> >
> > The claim that “Under partial observable environments, it is natural, and actually a must, for agents to do test time exploration’’ does not fully capture the problem. As stated in the Introduction (Line 38), an environment contains observable information (surface-level feedback) and implicit rules (deep-level factors). The former can be directly obtained through interaction, but the latter—while crucial for task success—must be inferred through deeper reasoning and hypothesis formation. Without explicit mechanisms that encourage the agent to explore these implicit rules, the agent often fails to discover them, leading to task failure (see Figure 1). Therefore, such exploration cannot be assumed to “naturally” happen.
> >
> > **(2) Our work focuses on online, real-time exploration, not offline environment mining.**
> >
> > While related works such as KnowAgent, KWM, and KnowSelf indeed address implicit rule extraction, they do so through offline knowledge mining—collecting trajectories or insights before test time. In contrast, our method performs rule discovery during a single execution episode, i.e., real-time exploration at test time, which is why we name it Test-Time Exploration.
> >
> > **(3) Our method implements the exploration behavior of the agent system.**
> >
> > Your statement that “the method is essentially training one model to guide another’’ refers to only one implementation. In our framework, the agent system—considered as a unified entity—carries out test-time exploration. Whether the actor and thinker are the same model, different models, or whether the thinker is trained or not are all instantiations of our general concept, not the essence of it.
> >
> > In summary, our method remains firmly grounded in the concept of test-time exploration, and to the best of our knowledge, it represents the first implementation that enables an agent system to simultaneously perform a task and actively uncover implicit environmental knowledge during execution.
> >
> > **Q3: Requirement of collecting successful trajectories.**
> >
> > Thank you for raising this point. In practice, obtaining a successful trajectory is not particularly difficult. As long as a task provides an outcome reward, one can apply standard reinforcement learning methods (e.g., vanilla GRPO) to gradually learn the task and the underlying environment dynamics until the model is capable of producing successful trajectories for most task instances.
> >
> > In fact, the successful trajectories used in our experiments are not those provided by the dataset. Instead, they were collected from an actor model trained via RL. The performance of this model is shown in Table 1 as Qwen2.5-Actor (No-Thinker) (second row from the bottom). We trained this actor solely using outcome rewards, and then used it to sample the successful trajectories.
> >
> > In reality, the primary bottleneck for scaling up our approach is not trajectory collection, but rather the scale and diversity of environments and tasks. Designing, selecting, and deploying new environments remains costly. Whether future research can provide sufficiently diverse environments, along with unified deployment pipelines, is likely to play a more important role in enabling large-scale applications of our method.

---

> > > ### Author Response · Authors · 2025-11-21
> > >
> > > **Q4: Benchmark Selection — Why not Webshop or more complex scenarios?**
> > >
> > > (1) **Our benchmark selection is guided primarily by whether a task provides an environment with meaningful exploration value.** Since our method focuses on improving an agent’s environmental exploration capabilities, we apply two core criteria when choosing evaluation tasks:
> > >
> > > a. The environment must contain implicit rules that meaningfully influence task success but cannot be directly observed.
> > >
> > > b. The agent must have sufficient interaction budget (interaction steps) to explore those rules before the task terminates.
> > >
> > > The benchmarks we selected (Alfworld, Sciworld, etc.) satisfy these requirements. Take Alfworld as an example:
> > >
> > > a. The environment contains hidden rules such as “an agent cannot hold two objects simultaneously” or “interacting with certain objects requires specific actions at specific locations.” These rules must be inferred from interaction history rather than being explicitly provided.
> > >
> > > b. Failed actions return a “Nothing happens” response, signaling to the agent that its intended operation did not succeed. Crucially, the environment does not reset after a failed attempt, allowing the agent to try multiple alternatives.
> > > Under these conditions, agents can meaningfully explore for dozens of steps before ultimately solving the task.
> > >
> > > (2) **Webshop is not suitable for evaluating test-time exploration.** Webshop essentially simulates a navigable online shopping interface in which all relevant information is explicitly provided. There are almost no implicit rules to be discovered through exploration. Additionally, both the action space and the typical episode length are extremely limited: the agent can only perform search and click, and the task structure follows a simple sequence—keyword search, product browsing, and a “buy” click. Once the agent clicks the “buy” button, the episode ends immediately, leaving no opportunity for extended or iterative exploration. For these reasons, we argue that Webshop is not an appropriate benchmark for evaluating test-time exploration capabilities.
> > >
> > > (3) **Future work will incorporate more complex scenarios.** In this paper, we include two in-domain tasks (Alfworld, Sciworld) and three out-of-domain tasks (BabyAI, Jericho, PDDL) as our evaluation benchmarks. In the future, we plan to explore more complex environments, such as GUI agent settings involving newly apps or functionalities, where exploration is more critical.
> > >
> > > **Q5: Why we use fixed-step triggering in our experiments.**
> > >
> > > Determining when a model should trigger deep thinking and how to perform high-quality deep thinking are both important yet challenging problems. The former relates to scheduling abilities, while the latter concerns a model’s deep reasoning capability.
> > >
> > > In this work, we focus primarily on the second problem—improving the quality of deep thinking. To control variables and isolate the effect of deep-thinking quality, we adopt a fixed-step triggering scheme.
> > >
> > > We agree that adaptive triggering could improve execution efficiency, and prior work (e.g., *KnowSelf*) attempts to train models to switch between “fast thinking,” “slow thinking,” and “knowledge thinking” modes. However, this adaptivity remains difficult for current baseline models: without explicit training in scheduling, prompt-based heuristics alone are typically insufficient for learning when to trigger deep thinking.
> > >
> > > To verify this, we added an experiment where the Actor and Thinker prompts were merged into a single prompt instructing the model to adaptively switch reasoning modes. Results show that this purely prompt-based adaptive approach often performs worse than simply using fixed thinking intervals.
> > >
> > > | Actor        | Thinker            | Alfworld | Sciworld | BabyAI | Jericho | PDDL  | Mean  |
> > > |--------------|--------------------|----------|----------|--------|---------|-------|--------|
> > > | Qwen2.5-7B   | No                 | 78.92    | 33.38    | 49.29  | 7.29    | 23.09 | 38.39 |
> > > |              | Self (Fixed Step)  | 80.78    | 54.41    | 67.43  | 16.76   | 17.71 | **47.42** |
> > > |              | Self (Auto)        | 46.89    | 48.12    | 51.77  | 11.73   | 33.35 | 38.37 |

---

> > > > ### Author Response · Authors · 2025-11-26
> > > >
> > > > Dear Reviewer aoxx,
> > > >
> > > > We sincerely thank you for your constructive comments and valuable suggestions! We have discussed these issues and look forward to your feedback. If you have any further questions, we would be happy to continue the discussion.
> > > >
> > > > Thank you again!
> > > >
> > > > Authors

---

### Official Review · Reviewer_7Rsq · 2025-11-01

**Soundness:** 2
**Presentation:** 3
**Contribution:** 2
**Rating:** 4
**Confidence:** 3

**Summary:**

The paper introduces TTExplore, which augments an acting agent with a specialized “thinker” model trained (via a mix of distillation and RL) to infer hidden environment rules and replan mid-trajectory, and shows large process-score gains across multiple interactive benchmarks using relatively small (~7B/8B) actors. The approach is interesting and practically relevant, but several core pieces currently feel partially hand-engineered (for ex. heuristics in the thinker RL pipeline, fixed intervention frequency), and some claims would benefit from stronger baselines (prompted large-model thinker, adaptive triggering of thinker mode.)

**Strengths:**

* Separates an “actor” from a dedicated “thinker” that explicitly infers hidden environment rules and replans at test time, which is a meaningful and novel framing for agent improvement (not just bigger models or more fine-tuning).
* Strong empirical gains: Consistently boosts performance across five interactive benchmarks, including out-of-domain settings, and lifts relatively small open models (7B/8B) to or near the level of much larger / heavily tuned baselines.
* The ablations are tested clearly along the distill, RL axes and has sufficient experiments on different model famlies and envs.

**Weaknesses:**

* The thinker training pipeline relies on several hand-crafted heuristics (trajectory segmentation, difficulty filtering, binary short-horizon rewards with a frozen actor), so it’s unclear how much of the benefit is general vs. engineered for these benchmarks and how transferable it is to new domains.
* The method is positioned as “test-time exploration,” but parts of the pipeline resemble best-of-N sampling and selection; the paper does not compare against a strong baseline where a large model is simply prompted to generate multiple reflective plans/rule hypotheses and the best one is chosen. A simple baselin of Best-of-N without replanning would strengthen the claims.
* Claims that a distilled+RL thinker beats scale are somewhat overstated: in some settings a huge off-the-shelf model still does better, especially out-of-domain (see Qwen 7B), and the paper doesn’t fully explain when targeted training wins vs. when raw scale wins.
* The thinker is triggered on a fixed schedule (every n steps), not adaptively when the agent is actually stuck, and there’s limited discussion of compute overhead or cost/latency tradeoffs versus plain actor baselines. The results on 'thinking' on every 3 steps for larger models and 6 for smaller models is counterintuitive, any hypothesis from the authors would be helpful to explain this behavior.

* Results are reported with Process Score, which rewards incremental progress; the paper doesn’t clearly report actual task success/completion rates or failure cases where the thinker’s inferred “rules” are confidently wrong and actively harms behavior.

**Questions:**

See weaknesses.

---

> ### Author Response · Authors · 2025-11-21
>
> Dear reviewer,
>
> Thank you very much for your comments and professional advice. We would like to show the detailed responses as follows:
>
> ---
>
> **Q1. Regarding the concern that our training pipeline relies on several hand-crafted heuristics, leading to uncertain generality and transferability.**
>
> **(1) Why such a training design is necessary.**
>
> When the target that a model needs to learn is inherently difficult to evaluate, training becomes fundamentally challenging.
> In our setting, this *“hard-to-evaluate”* target is the *deep thinking results* within an agent system.
> Our multi-stage pipeline is specifically designed to convert this unstable and hard-to-assess signal into intuitive and stable rewards that can effectively guide learning.
>
> **(2) Generality and transferability of the method.**
>
> In practice, most components of our training pipeline are **highly general and easy to transfer across tasks**.
> Techniques such as **difficulty filtering** and **executing trajectories with a frozen actor** do not rely on task-specific assumptions.
>
> Even the most demanding step—*obtaining a successful trajectory with process scores*—does not fundamentally require golden paths or ground-truth intermediate rewards.
>
> As long as a task provides a **final outcome reward**, one can train an agent (e.g., using a standard GRPO algorithm) until it can reliably produce successful trajectories.
> Once a successful trajectory exists, a large model can be used to segment it into sub-tasks, thereby generating process scores even for environments that do not natively provide them.
>
> **(3) Clarification of our experimental setup.**
>
> Importantly, the *successful trajectories* used in our experiments are **not** from the original datasets;
> they are **collected from an Actor model trained via RL**.
> The performance of this model is reported in Table 1 as **“Qwen2.5-Actor (No Thinker)”** (second row from the bottom).
>
> For process scores:
>
> * Sciworld provides its own intermediate rewards.
> * Alfworld is scored using manually designed rules following the agentboard framework.
> * For tasks without any built-in process rewards, leveraging a large model to assign scores based on a successful trajectory is a **practical and effective** solution.
>
> **Q2. The execution pipeline resembles an N-sample best-of selection, yet no comparison with Best-of-N is provided.**
>
> Our contributions mainly lie in two aspects:
> **(1) the Test-Time Exploration (TTE) framework**, and **(2) a training method for the Thinker model**.
>
> Although our training process involves multiple sampling-and-evaluation iterations,
> our **inference-time procedure uses only a single sample**, identical to the step-budget constraints of all baselines.
> Thus, our method **does not** rely on Best-of-N resampling during testing, and the execution-time cost is fully aligned with the baselines.
>
> While the Thinker may consider multiple possible solution directions during deep thinking,
> the Actor performs actions **sequentially, without branching or backtracking**.
> This design is consistent with how humans explore real environments—tasks are irreversible and exploration must occur within a single action budget.
>
> Furthermore, our method is **orthogonal** to Best-of-N and can be combined with it.
> To validate this, we conducted an additional experiment with **N = 5**, using *Qwen2.5-7B-Instruct* as the Actor and our **Exp-Thinker** as the Thinker.
> The results show that even under the Best-of-N setup, **our method continues to yield clear advantages**.

---

> > ### Author Response · Authors · 2025-11-21
> >
> > | Method           | Alfworld Score | Alfworld Time(s) | Sciworld Score | Sciworld Time(s) | BabyAI Score | BabyAI Time(s) | Jericho Score | Jericho Time(s) | PDDL Score | PDDL Time(s) | Mean Score | Total Time (s) | Cost |
> > |-----------------|----------------|-----------------|----------------|-----------------|--------------|----------------|---------------|----------------|------------|--------------|------------|----------------|------|
> > | ReAct           | 78.9           | 71              | 33.4           | 95              | 49.3         | 91             | 7.3           | 53             | 23.1       | 79           | 38.4       | 389            | 1.0  |
> > | Reflexion (N times)      | 85.6           | 341             | 56.9           | 436             | 75.6         | 469            | 18.0          | 291            | 38.3       | 301          | 54.9       | 1838           | 4.7  |
> > | Best-of-N       | 96.3           | 357             | 70.5           | 443             | 84.2         | 467            | 26.7          | 295            | 54.0       | 362          | 66.3       | 1925           | 5.0  |
> > | TTE             | 91.9           | 111             | 75.9           | 116             | 69.4         | 130            | 19.2          | 99             | 21.4       | 104          | 55.5       | 559            | 1.4  |
> > | TTE+Best-of-N   | 99.4           | 589             | 91.1           | 606             | 85.3         | 687            | 31.5          | 507            | 43.8       | 519          | 70.2       | 2907           | 7.5  |
> >
> >
> > **Q3. Large, untrained models can sometimes outperform well-trained smaller models, especially on out-of-domain tasks.**
> >
> > Large models naturally possess broader and deeper reasoning capabilities due to their substantial parameter capacity. In deep thinking scenarios, it is therefore unsurprising that they can exhibit stronger performance. Our goal, however, is to train a smaller model so that it can match—or even surpass—larger models in most settings. Indeed, on in-domain tasks, our trained small model achieves this, as shown in Table 2.
> >
> > That said, its out-of-domain generalization—the case of Qwen2.5-7B compared to the 10× larger Qwen2.5-72B—does not yet fully reach the level of the much larger model. Nevertheless, we expect this gap to continue shrinking as we incorporate a more diverse set of training tasks.
> >
> >
> > **Q4. The method uses fixed-step triggering rather than adaptive triggering. There is limited discussion on computation cost, and it is unclear why larger models perform better with a 3-step interval while smaller models prefer a 6-step interval.**
> >
> > **(1) Why we use fixed-step triggering in our experiments**
> >
> > Determining **when** a model should trigger deep thinking and **how** it should perform high-quality deep thinking are both important yet challenging problems.
> > The former involves *scheduling ability*, while the latter concerns a model’s *deep reasoning capability*.
> >
> > In this work, we focus primarily on the second problem—**improving the quality of deep thinking**.
> > To isolate the effect of deep-thinking quality and control unnecessary variables, we adopt a **fixed-step triggering** scheme.
> >
> > We agree that **adaptive triggering** could potentially improve execution efficiency. Prior work (e.g., *KnowSelf*) attempts to train models to switch between “fast thinking,” “slow thinking,” and “knowledge thinking” modes. However, this adaptivity remains difficult for current baseline models:
> > without explicit training on scheduling, **prompt-based heuristics alone are insufficient** for reliably learning when to trigger deep thinking.
> >
> > To verify this, we conducted an additional experiment that **merged the Actor and Thinker prompts** into a single prompt, instructing the model to adaptively switch reasoning modes.
> > The results show that this purely prompt-based adaptive approach often performs **worse** than using fixed thinking intervals.
> >
> > | Actor        | Thinker            | Alfworld | Sciworld | BabyAI | Jericho | PDDL  | Mean  |
> > |--------------|--------------------|----------|----------|--------|---------|-------|--------|
> > | Qwen2.5-7B   | No                 | 78.92    | 33.38    | 49.29  | 7.29    | 23.09 | 38.39 |
> > |              | Self (Fixed Step)  | 80.78    | 54.41    | 67.43  | 16.76   | 17.71 | **47.42** |
> > |              | Self (Auto)        | 46.89    | 48.12    | 51.77  | 11.73   | 33.35 | 38.37 |

---

> > > ### Author Response · Authors · 2025-11-21
> > >
> > > **(2) Discussion on computation cost and latency**
> > >
> > > Our inference setup uses **vLLM with parallel decoding** across samples and **parallel environment execution**.
> > > In our tasks, the Actor’s maximum output length is **128 tokens**, and the Thinker’s output is capped at **512 tokens**.
> > >
> > > Using *Qwen2.5-7B-Instruct* with our **Exp-Thinker**, we measured the execution time across tasks.
> > > On average, our TTE framework requires **~1.4×** the execution time of standard ReAct.
> > > This overhead is expected due to the additional deep-thinking steps.
> > >
> > > At the same time, our method is **fully compatible with Best-of-N**, enabling users to trade additional compute for higher success rates when desired.
> > >
> > > The table can be seen in Q2.
> > >
> > > **(3) Why larger models prefer shorter intervals and smaller models prefer longer intervals.**
> > >
> > > As discussed in Section 4.5, we hypothesize that **deep thinking quality directly affects the Actor’s performance**. High-quality thinking can guide the Actor toward correct actions more quickly, whereas frequent low-quality thinking may clutter the context, lower the signal-to-noise ratio, and ultimately harm decision-making.
> > >
> > > Larger models generally produce higher-quality deep thinking more consistently, so **more frequent triggering (e.g., every 3 steps)** is beneficial. Smaller models produce noisier thoughts, so **less frequent triggering (e.g., every 6 steps)** helps maintain a cleaner context, leading to better overall performance.
> > >
> > > **Q5: Missing task success rate and example of a failure caused by incorrect rule inference.**
> > >
> > > Thank you for the suggestion. Due to space limitations, we were unable to include task success rates in the main paper. We have now added the success rates corresponding to the results in Table 1 (shown below).
> > >
> > > | Actor        | Thinker      | Alfworld (succ / proc) | Sciworld (succ / proc) | BabyAI (succ / proc) | Jericho (succ / proc) | PDDL (succ / proc) | Mean  |
> > > |--------------|--------------|-------------------------|--------------------------|------------------------|-------------------------|----------------------|--------|
> > > | LLaMA3-8B    | No           | 7.46 / 32.15           | 13.33 / 20.31           | 26.78 / 42.44         | 0.00 / 14.23           | 10.00 / 29.94        | 27.81 |
> > > |              | Exp-Thinker  | 32.08 / 63.49          | 44.44 / 55.43           | 41.96 / 58.15         | 5.00 / 26.31           | 16.66 / 30.06        | 46.69 |
> > > | Qwen2.5-7B   | No           | 68.65 / 80.03          | 33.33 / 39.82           | 37.50 / 50.16         | 0.00 / 8.95            | 18.33 / 25.38        | 40.87 |
> > > |              | Exp-Thinker  | 90.29 / 94.96          | 65.55 / 72.69           | 50.00 / 63.12         | 5.00 / 19.94           | 8.33 / 20.87         | 54.32 |
> > > | LLaMA3-Actor | No           | 68.66 / 78.92          | 45.56 / 54.87           | 23.21 / 35.55         | 0.00 / 15.98           | 5.00 / 20.19         | 41.10 |
> > > |              | Exp-Thinker  | 77.61 / 87.75          | 51.11 / 63.14           | 35.71 / 46.41         | 0.00 / 17.11           | 10.00 / 32.11        | 49.30 |
> > > | Qwen2.5-Actor| No           | 94.77 / 97.76          | 77.77 / 83.07           | 40.17 / 50.62         | 0.00 / 17.44           | 15.00 / 29.76        | 55.73 |
> > > |              | Exp-Thinker  | 97.76 / 98.50          | 74.44 / 82.27           | 48.21 / 60.25         | 0.00 / 17.64           | 11.66 / 31.43        | 58.02 |
> > >
> > > We also provide an example of a failed task. In this scenario, the goal is to inspect a bowl using a desk lamp. During execution, however, the Actor becomes fixated on the lamp and repeatedly attempts to interact with it, failing to make progress toward the actual objective. The Thinker’ deep-thinking traces do reason about the lamp’s position, the bowl’s position, and illumination factors, but never uncover the key implicit rule: the bowl must be **picked up** before it can be inspected under the lamp. Because the agent does not discover this rule, it ultimately exceeds the maximum step limit and fails the task.

---

> > > > ### Author Response · Authors · 2025-11-21
> > > >
> > > > ```
> > > > Task: look at bowl under the desklamp.
> > > > Init Observation: You are in the middle of a room. Looking quickly around you, you see a bed 1, a desk 2, a desk 1, a drawer 6, a drawer 5, a drawer 4, a drawer 3, a drawer 2, a drawer 1, a garbagecan 1, a laundryhamper 1, a safe 1, a shelf 6, a shelf 5, a shelf 4, a shelf 3, a shelf 2, and a shelf 1.
> > > > >go to desk 1
> > > > On the desk 1, you see a desklamp 1, a keychain 3, a mug 2, a pen 2, a pen 1, and a pencil 2.
> > > > >go to desk 2
> > > > On the desk 2, you see a alarmclock 1, a bowl 1, and a cd 3.
> > > > >go to desk 1
> > > > On the desk 1, you see a desklamp 1, a keychain 3, a mug 2, a pen 2, a pen 1, and a pencil 2.
> > > > >use desklamp 1
> > > > You turn on the desklamp 1.
> > > > >look at bowl 1 under desklamp 1
> > > > Nothing happens.
> > > > >use desklamp 1
> > > > You turn on the desklamp 1.
> > > > DEEPTHINK: The history trajectory seems to be following a logical sequence of actions, but the latest step is confusing. Specifically, when the agent attempts to "look at bowl 1 under desklamp 1," nothing happens. This could be due to a few reasons:
> > > > 1. **Visibility Issue**: The desklamp might not be illuminating the area under it sufficiently, or the bowl might be obstructed by something else.
> > > > 2. **Receptacle Position**: The bowl might not be directly under the desklamp, or there might be an object in between.
> > > > 3. **Hidden Rule**: There might be a rule that the bowl cannot be seen under the desklamp, perhaps because it is not a common action or because the environment has a specific layout that prevents this.
> > > > To clarify, I will first check if the desklamp is illuminating the area properly and then check the position of the bowl.
> > > > >use desklamp 1
> > > > You turn on the desklamp 1.
> > > > >look at bowl 1 under desklamp 1
> > > > Nothing happens.
> > > > >go to desk 2
> > > > On the desk 2, you see a alarmclock 1, a bowl 1, and a cd 3.
> > > > >use desklamp 1
> > > > Nothing happens.
> > > > >look at bowl 1 under desklamp 1
> > > > Nothing happens.
> > > > >go to desk 1
> > > > On the desk 1, you see a desklamp 1, a keychain 3, a mug 2, a pen 2, a pen 1, and a pencil 2.
> > > > DEEPTHINK: The task is to "look at bowl 1 under desklamp 1." The history trajectory shows that the desklamp 1 was used multiple times, but the action "look at bowl 1 under desklamp 1" did not produce any feedback, which is unusual. Here are the possible reasons for this:
> > > > 1. **Desklamp Placement**: The desklamp 1 might not be positioned in a way that allows it to illuminate the area under it. This could be due to its height or angle.
> > > > 2. **Bowl Position**: The bowl 1 might not be directly under the desklamp 1. It could be placed elsewhere on the desk or not in a position that would be illuminated by the desklamp.
> > > > 3. **Light Intensity**: The desklamp 1 might not be providing enough light to clearly see the area under it, especially if the bowl is in a shadow or the light is not directed downwards.
> > > > To verify these hypotheses, I will first check the position of the desklamp and bowl, and then attempt to move the bowl to a position under the desklamp.
> > > > >use desklamp 1
> > > > You turn on the desklamp 1.
> > > > >look at bowl 1 under desklamp 1
> > > > Nothing happens.
> > > > >go to desk 2
> > > > On the desk 2, you see a alarmclock 1, a bowl 1, and a cd 3.
> > > > >use desklamp 1
> > > > Nothing happens.
> > > > >look at bowl 1 under desklamp 1
> > > > Nothing happens.
> > > > >go to desk 1
> > > > On the desk 1, you see a desklamp 1, a keychain 3, a mug 2, a pen 2, a pen 1, and a pencil 2.
> > > > DEEPTHINK: The task is to "look at bowl 1 under desklamp 1." The history trajectory shows that the agent has attempted to perform this action multiple times, but "Nothing happens" each time. This suggests that the environment might not be interpreting the phrase "under desklamp 1" correctly or there might be an implicit rule that the bowl is not actually under the desklamp.
> > > > Let's consider the following hypotheses:
> > > > 1. The bowl is not physically under the desklamp but is simply on the desk.
> > > > 2. The desklamp 1 is not correctly associated with desk 2, even though the agent is at desk 2 when trying to use it.
> > > > To verify these hypotheses, we need to:
> > > > - Confirm the location of the bowl and desklamp.
> > > > - Check if the desklamp can be used at different desks.
> > > > Let's first check the current state of the environment:
> > > > - The bowl 1 is on desk 2.
> > > > - The desklamp 1 is on desk 1.
> > > > Given this information, the bowl is not under the desklamp, but on the desk. The desklamp is on a different desk than where the bowl is located.
> > > > Next, we need to check if the desklamp can be used at different desks. Let's try to use the desklamp 1 at desk 2 to see if it works there.
> > > > >go to desk 2
> > > > On the desk 2, you see a alarmclock 1, a bowl 1, and a cd 3.
> > > > >use desklamp 1
> > > > Nothing happens.
> > > > >look at bowl 1 on desk 2
> > > > Nothing happens.
> > > > >use desklamp 1
> > > > Nothing happens.
> > > > >look at bowl 1 on desk 2
> > > > Nothing happens.
> > > > …
> > > > [Task Fail]
> > > > ```
> > > > ---
> > > > Finally, we would like to thank you again for taking the time to review our manuscript.

---

> > > > > ### Author Response · Authors · 2025-11-26
> > > > >
> > > > > Dear Reviewer 7Rsq,
> > > > >
> > > > > We sincerely thank you for your constructive comments and valuable suggestions! We have discussed these issues and look forward to your feedback. If you have any further questions, we would be happy to continue the discussion.
> > > > >
> > > > > Thank you again!
> > > > >
> > > > > Authors

---

### Official Review · Reviewer_C4Hx · 2025-11-01

**Soundness:** 3
**Presentation:** 3
**Contribution:** 3
**Rating:** 6
**Confidence:** 4

**Summary:**

This paper proposes TTExplore, a framework for improving LLM-based agents in environments with implicit, unobservable rules. The method separates reasoning and acting into a Thinker–Actor setup, where the Thinker infers hidden constraints at test time to guide the Actor. The agent formalize reward functions to train the LLM thinkers and experiments on five text-based embodied benchmarks show consistent improvements.

**Strengths:**

- The paper focuses on an important and interesting problem: enabling LLM-based agents to handle implicit environmental rules at test time.
- The proposed Thinker–Actor separation is clearly motivated, where the Thinker discovers hidden constraints and replans to guide a lower-level Actor, and the paper is well-organized, easy to follow,
- The paper includes comprehensive experiments across multiple benchmarks, with ablation studies and analysis that convincingly support the claims.

**Weaknesses:**

- The approach is largely a combination of existing ideas and algorithms. The overall pipeline of letting LLMs proposing candidates, letting a low-level actor act with the outcomes, and using selected one to refine the model resembles many previous frameworks [1].
---
[1] Lee, Dongjun, et al. "Learning to contextualize web pages for enhanced decision making by LLM agents." arXiv preprint arXiv:2503.10689 (2025).

**Questions:**

- Is the reward for the Thinker binary (0/1)? Have the authors tried using a progression-based reward to provide denser signals?

---

> ### Author Response · Authors · 2025-11-21
>
> Dear reviewer,
>
> Thank you very much for your comments and professional advice. We would like to show the detailed responses as follows:
>
> ---
>
> **Q1: The proposed method appears to be a combination of existing ideas and similar to prior frameworks.**
>
> Our contributions consist of two key innovations:
>
> (1) We introduce a **test-time exploration framework** that incorporates a “deep thinking” phase. This additional reasoning stage enables the agent to uncover latent knowledge or rules in the environment, improving its ability to generalize.
>
> (2) We propose a **training paradigm that enhances the quality of deep thinking.** Instead of directly evaluating the inherently hard-to-measure *deep thoughts*, we transform this into an evaluation of the low-level actor’s execution results, which are stable and easy to assess.
>
> We carefully examined the paper you recommended, *“Learning to contextualize web pages for enhanced decision making by LLM agents”*. Indeed, its training strategy and ours share a similarity: both convert a difficult-to-evaluate signal into an easier-to-evaluate one. In that work, the hard-to-evaluate signal is the *processed environment observation*, while in ours it is the *deep thinking result*.
>
> However, the underlying mechanisms differ substantially. The referenced paper trains the model via supervised fine-tuning on selected correct outputs, essentially resembling a rejection sampling process. In contrast, **our method integrates GRPO and updates the model through reinforcement learning**, enabling gradual policy improvement driven by performance feedback rather than supervised selection.
>
> **Q2: Is the reward for the thinker binary (0/1)? Have you tried denser signals?**
>
> Yes, our current training setup uses a **binary reward**. Specifically, we assign a reward of **1** if, after receiving the deep-thinking output, the actor achieves any increase in process score within a fixed number of steps; otherwise, the reward is **0**.
>
> We appreciate your suggestions, and we agree that using denser signals is theoretically beneficial. For example, the *time (number of steps)* required to achieve the first score improvement can serve as a finer-grained metric. Motivated by this, we conducted an additional experiment using a more granular reward.
>
> In this experiment, we introduced a **step penalty of −0.05** for each generation step taken *before* the first score increase. The longer it takes for the actor to achieve improvement, the more cumulative penalty is applied. Using this reward, we trained a new model, **Thinker-v2**. For a fair comparison, we also retrained our original thinker under the original 0/1 reward, naming it **Thinker-v1**.
>
> Both models were trained on the same datasets, with identical hyperparameters, and for 400 steps. Training curves show that both converge stably. We then evaluated Thinker-v1 and Thinker-v2 using two untrained actors—**Qwen2.5-7B** and **Llama3-8B**—within our test-time exploration framework. Results indicate that denser rewards help more on *in-domain* data, while the effect is minimal on *out-of-domain* tasks.
>
> | Actor       | Thinker      | Alfworld | Sciworld | BabyAI | Jericho | PDDL  | Mean  |
> |------------|-------------|----------|----------|--------|---------|-------|-------|
> | Llama3-8B  | No          | 34.39    | 25.32    | 37.01  | 18.13   | 27.78 | 28.53 |
> |            | Thinker-v1  | 49.32    | 69.39    | 64.91  | 25.10   | 30.19 | 47.78 |
> |            | Thinker-v2  | **53.86**    | **71.69**    | 63.10  | 20.28   | 27.41 | 47.27 |
> | Qwen2.5-7B | No          | 78.92    | 33.38    | 49.29  | 7.29    | 23.09 | 38.39 |
> |            | Thinker-v1  | 88.62    | 68.52    | 64.46  | 17.74   | 20.92 | 52.05 |
> |            | Thinker-v2  | **91.04**    | **78.20**    | 66.25  | 16.38   | 22.60 | 54.89 |
>
>
> We plan to include this as an ablation study in the main paper or appendix.
>
> ---
>
> Finally, we would like to thank you again for taking the time to review our manuscript.

---

> > ### Author Response · Authors · 2025-11-26
> >
> > Dear Reviewer C4Hx,
> >
> > We sincerely thank you for your constructive comments and valuable suggestions! We have discussed these issues and look forward to your feedback. If you have any further questions, we would be happy to continue the discussion.
> >
> > Thank you again!
> >
> > Authors

---

### Author Response · Authors · 2025-11-25

Dear reviewers,

We sincerely thank you for your constructive comments and valuable suggestions! We have discussed these issues and look forward to your feedback. If you have any further questions, we would be happy to continue the discussion.

Thank you again!

Authors

---

### Author Response · Authors · 2025-12-02
**Brief Summary**

This work investigates the generalization challenge of agents operating in **unknown environments**. We identify that a significant portion of agent failures stems from a lack of understanding of the **implicit rules** governing such environments.

To address this, we made the following contributions:

(1) A **Test-Time Exploration (TTExplore)** framework that introduces an explicit *Thinker* role to guide real-time environmental exploration during interaction.

(2) Additionally, we devise a **reinforcement learning training pipeline based on task decomposition and indirect reward**, converting a difficult-to-assess training objective into quantifiable, stable metrics. Using this pipeline, we train a *Thinker Model* which—when deployed in the TTExplore framework—significantly improves the task performance of the base *Actors*.

Here are the key strengths highlighted by the reviewers:
- The paper focuses on **an important and interesting problem**: enabling LLM-based agents to handle implicit environmental rules at test time. *(By Reviewer C4Hx)*

- Introduces **a novel and well-motivated Thinker-Actor framework** that enables agents to infer hidden environmental rules and replan at test time, addressing a crucial challenge in adaptive reasoning. *(By Reviewers C4Hx, 7Rsq, and UZ3W)*

- The paper demonstrates **solid methodological rigor** through a carefully constructed reinforcement learning pipeline that tackles the instability of sparse rewards via task decomposition and difficulty filtering. *(By Reviewer UZ3W)*

- **Strong empirical gains and comprehensive experiments**, demonstrating substantial improvements across multiple interactive benchmarks and model families, lifting smaller open models to the level of much larger or heavily tuned baselines. *(By Reviewers C4Hx, 7Rsq and UZ3W)*

- The paper is **well written and structured**, with clear motivation, diagrams, and examples that effectively illustrate the challenges of implicit rule reasoning. *(By Reviewers C4Hx, aoxx and UZ3W)*

Summary of key issues addressed：

- **Clarification of Novelty.**
We clarified the core innovations of our work and articulated how our approach fundamentally differs from prior methods that rely on offline extraction of environment patterns. *(Addressed to Reviewers C4Hx and aoxx)*

- **Transferability of the Training Method.**
We explained that our training strategy is inherently transferable and provided further analysis supporting its generalization potential. *(Addressed to Reviewers 7Rsq and aoxx)*

- **Impact of Denser Reward Signals.**
We added experiments analyzing the effect of using denser reward signals during training. Results show that replacing binary rewards (0/1) with more dense signals accelerates convergence and improves in-domain performance at equal training budgets, while out-of-domain performance remains largely unchanged. *(Addressed to Reviewer C4Hx)*

- **Distinction from Test-Time Scaling Methods.**
We clarified the differences between our method and traditional test-time scaling approaches such as Reflexion and Best-of-N. We further added comparative experiments and reported time-cost statistics, demonstrating the efficiency advantage of our method. *(Addressed to Reviewers 7Rsq and UZ3W)*

- **Rationale for Using a Fixed Trigger for Deep Thinking.**
We justified our choice of a fixed trigger depth in the experimental setup, emphasizing that our goal is to improve the quality of deep thinking rather than to optimize the triggering policy. Additional experiments show that prompting the model to decide its own thinking frequency does not outperform fixed-frequency triggering. *(Addressed to Reviewers 7Rsq and aoxx)*

- **Choice of Evaluation Benchmarks and Discussion of Multimodal Extension.**
We detailed the criteria for selecting our benchmark environments—specifically, environments where exploration is meaningful. We also explained why WebShop was not suitable. In addition, we discussed the applicability of our method to multimodal settings, highlighting both their commonalities and the challenge of potential context overflows during direct transfer. *(Addressed to Reviewers aoxx and UZ3W)*

- **Comparison to Additional Offline-Knowledge Methods.**
We extended our related-work discussion to include a broader set of methods that leverage offline exploration to acquire environment knowledge, such as Expel, KnowAgent, KWM, and KnowSelf. *(Addressed to Reviewers aoxx and UZ3W)*


- **Effect of the Number of Thinking Nodes Per Trajectory on Training Stability.**
We added experiments analyzing how the number of “thinking nodes” in a single trajectory influences training stability. Results indicate that fewer thinking nodes per trajectory lead to more stable training and better test performance. When the number reaches six (i.e., six thinking nodes sharing a single outcome reward), training becomes unstable and collapses. *(Addressed to Reviewer UZ3W)*

---

### Meta-Review · Area_Chair_rCfx · 2026-01-09

**Summary:**

This paper introduces Test-Time Exploration (TTExplore), a framework designed to enhance large language model (LLM)-based agents operating in environments with implicit rules—that is, hidden constraints that cannot be directly observed and must be inferred through interaction. TTExplore introduces a two-role architecture: an actor that performs ReAct-style actions, and a thinker that periodically conducts “deep thinking” to infer unobservable environmental rules from the interaction history and guide the actor’s future decisions.

To train the thinker, the authors develop a stable reinforcement learning pipeline that mitigates the challenges of sparse rewards. Evaluations on five text-based embodied benchmarks (ALFWorld, SciWorld, BabyAI, Jericho, and PDDL) show that TTExplore improves agent performance. The approach also enhances well-trained agents on out-of-domain tasks, demonstrating strong generalization and complementary benefits to traditional SFT/RL methods.

The primary concerns from reviewers were:
1. Combination of ideas from existing work
2. Reliance on hand crafted heuristics.
3. Does not compare to strong baselines like best-of-N or Reflexion
4. Not clear when the method works well.
5. limited discussion of compute overhead or cost/latency tradeoffs versus plain actor baselines.
6. requires presence of ground truth trajectory
7. Not clear if it falls into bucket of test time scaling.
8. Limited theoretical grounding.
9. Broader evaluation.

**Reviewer Concerns:**

Simply saying something is a combination of existing methods / limited novelty is not a sufficient criticism, so I will dismiss these. The authors also clarify this. They also add.

Explanations of the transferability of the Training Method.
Experiments quantifying impact of Denser Reward Signals.
Distinction from Test-Time Scaling Methods.
Choice of Evaluation Benchmarks and Discussion of Multimodal Extension.
Related work expansion to Additional Offline-Knowledge Methods.
Additional experiment on Effect of the Number of Thinking Nodes Per Trajectory on Training Stability.

Overall the additions do help address some of the concerns of the reviewers, but I think the fundamental challenges of the method being somewhat convoluted, relying on hand-written heuristics and not comparing to enough baseline methods and evaluation environments remain. I didn't find the explanations of why test-time scaling methods should not be considered very convincing. These should be rectified before acceptance.

**Reviewer Scores:**

C4Hx would likely remain at a 6
7Rsq would likely increase to a 5.
aoxx would likely go up to a 4 given the clarifications/experiments.
UZ3W would likely stay at a 6, maybe go up to a 7 given concerns on evaluation/baselines and scalability.

---

### Decision · Program_Chairs · 2026-01-26

Reject